# REDUCING CONTEXTUAL STOCHASTIC BILEVEL OPTIMIZATION VIA STRUCTURED FUNCTION APPROXIMATION

**Maxime Bouscary**
Operations Research Center
Massachusetts Institute of Technology
mbscry@mit.edu

**Jiawei Zhang**
Department of Computer Sciences
University of Wisconsin-Madison
jzhang2924@wisc.edu

**Saurabh Amin**
Laboratory for Information & Decision Systems
Massachusetts Institute of Technology
amins@mit.edu

## ABSTRACT

Contextual Stochastic Bilevel Optimization (CSBO) extends standard stochastic bilevel optimization (SBO) by incorporating context-dependent lower-level problems. CSBO problems are generally intractable since existing methods require solving a distinct lower-level problem for each sampled context, resulting in prohibitive sample and computational complexity, in addition to relying on impractical conditional sampling oracles. We propose a reduction framework that approximates the lower-level solutions using expressive basis functions, thereby decoupling the lower-level dependence on context and transforming CSBO into a standard SBO problem solvable using only joint samples from the context and noise distribution. First, we show that this reduction preserves hypergradient accuracy and yields an $\epsilon$-stationary solution to CSBO. Then, we relate the sample complexity of the reduced problem to simple metrics of the basis. This establishes sufficient criteria for a basis to yield $\epsilon$-stationary solutions with a near-optimal complexity of $\widetilde{\mathcal{O}}(\epsilon^{-3})$, matching the best-known rate for standard SBO up to logarithmic factors. Moreover, we show that Chebyshev polynomials provide a concrete and efficient choice of basis that satisfies these criteria for a broad class of problems. Empirical results on inverse and hyperparameter optimization demonstrate that our approach outperforms CSBO baselines in convergence, sample efficiency, and memory usage.[1]

## 1 INTRODUCTION

Many real-world optimization tasks involve solving a bilevel problem where the lower-level solution depends on the upper-level uncertainty. Such tasks can be framed as Contextual Stochastic Bilevel Optimization (CSBO) problems:

$$\min_{x \in \mathbb{R}^{d_x}} \quad F(x) \triangleq \mathbb{E}_{(\xi,\eta) \sim \mathbb{P}_{(\xi,\eta)}} \left[ f(x, y^\star(x,\xi), \xi, \eta) \right] \qquad \text{(CSBO)}$$

$$\text{s.t.} \quad y^\star(x,\xi) = \arg\min_{y \in \mathbb{R}^{d_y}} \mathbb{E}_{\eta \sim \mathbb{P}_{\eta|\xi}} \left[ g(x, y, \xi, \eta) \right], \quad \forall x \in \mathbb{R}^{d_x}, \xi \in \Xi$$

where $\mathbb{P}_{(\xi,\eta)}$ denotes the joint distribution of $\xi \in \mathbb{R}^{d_\xi}$ and $\eta \in \mathbb{R}^{d_\eta}$, and $\mathbb{P}_{\eta|\xi}$ denotes the conditional distribution of $\eta$ given $\xi$. Here, $\Xi \subseteq \mathbb{R}^{d_\xi}$ is the support of the random variable $\xi$. The dimension of the upper and lower-level variables $x$ and $y$ are $d_x$ and $d_y$, respectively. The function $f(x, y, \xi, \eta)$ can be nonconvex in $x$, but $g(x, y, \xi, \eta)$ must be strongly convex with respect to $y$ for any given $x, \xi$, and $\eta$, ensuring that the lower-level solution $y^\star(x,\xi)$ is uniquely defined. Our goal is to develop a

---

[1]The code is publicly available at: https://github.com/mbscry/CSBO-reduction.

reduction framework that enables the use of existing stochastic bilevel optimization algorithms to compute an $\epsilon$-stationary point $x^\star$, i.e. a point satisfying $\mathbb{E}\|\nabla F(x^\star)\|^2 \leq \epsilon^2$ where the expectation is taken over the randomness of the algorithm producing $x^\star$.

Stochastic Bilevel Optimization (SBO) is a special case of CSBO where the lower level is independent of $\xi$, in which case the solution to the lower level $y^\star$ depends only on $x$. In this setting, methods like stocBiO (Ji et al., 2021) and STABLE (Chen et al., 2022) achieve a sample complexity in $\tilde{\mathcal{O}}(\epsilon^{-4})$. With the use of variance reduction techniques or momentum-based estimators, sample complexities improve to $\tilde{\mathcal{O}}(\epsilon^{-3})$ (Guo et al., 2021; Khanduri et al., 2021; Yang et al., 2023) or even $\mathcal{O}(\epsilon^{-3})$ (Chu et al., 2024). These results match the lower bound for *single-level* stochastic optimization (Fang et al., 2018; Cutkosky & Orabona, 2019; Arjevani et al., 2022), suggesting that SBO can be solved efficiently in theory. This raises a natural question: *can we achieve similarly efficient solutions in the more general CSBO setting?*

A naïve adaptation of SBO algorithms to CSBO computes $\epsilon$-stationary solutions by solving $\Theta\left(\epsilon^{-2}\right)$ lower-level problems at each iterate $x_t$, each corresponding to a different $\xi$, to estimate the hypergradient $\nabla F(x_t)$. Specifically, using this naive approach to solve CSBO problems, the sample complexity increases from $\tilde{\mathcal{O}}(\epsilon^{-4})$ to $\tilde{\mathcal{O}}(\epsilon^{-6})$, and from $\tilde{\mathcal{O}}(\epsilon^{-3})$ to $\tilde{\mathcal{O}}(\epsilon^{-5})$ with variance reduction methods. Using multilevel Monte Carlo methods, Hu et al. (2023b) reduces the sample complexity to $\tilde{\mathcal{O}}(\epsilon^{-4})$. However, the practical efficiency of their algorithm relies on the existence and knowledge of a reference point $y_0(x)$ that is close to $y^\star(x, \xi)$ for all $\xi$, which may not hold in practice. Alternatively, one can approximate (CSBO) by partitioning the context space $\Xi$ into $N$ sub-regions and form a SBO problem with $N$ subproblems. While this approximation becomes more accurate as $N$ increases, it introduces redundant computation across adjacent regions, resulting in slow convergence. In the specific case where $\Xi$ is discrete and of finite cardinality $m$, Guo et al. (2021) achieves a sample complexity in $\tilde{\mathcal{O}}(m\epsilon^{-3})$. As such, their approach is not amenable to settings where $\xi$ is continuous. In addition, all of the above algorithms require a sampling oracle from the conditional distribution $\mathbb{P}_{\eta|\xi}$ for any fixed $\xi$, which can be impractical. For example, in inverse optimization or federated learning, although the joint distribution is often easily observable, the offline nature of the data or privacy constraints can prohibit conditional sampling. Our framework circumvents this limitation by needing only i.i.d. samples from the joint distribution, making it broadly applicable to realistic CSBO problems. This issue does not arise in standard SBO, where the lower-level distribution does not depend on $\xi$.

Despite recent progress, a significant gap remains between the complexity and practicality of SBO and CSBO methods. Specifically, methods for solving CSBO suffer from the following drawbacks:

1. Sample inefficiency: existing CSBO algorithms incur at least a $\tilde{\mathcal{O}}(\epsilon^{-1})$ factor relative to the optimal $\mathcal{O}(\epsilon^{-3})$ rate for SBO.

2. Oracle dependence: reliance on an often-unavailable oracle for sampling from the conditional distributions $\mathbb{P}_{\eta|\xi}$ for all fixed $\xi$.

We show that, under mild conditions on $g$ and $\mathbb{P}_\xi$, it is in fact possible to achieve a $\tilde{\mathcal{O}}(\epsilon^{-3})$ complexity for CSBO – matching that of SBO up to logarithmic factors – using only samples from the joint distribution $\mathbb{P}_{(\xi,\eta)}$.

The main challenge in solving (CSBO) lies in the coupling between $x$ and $\xi$ through the lower level solutions $y^\star(x, \xi)$. To address it, we propose to decouple this dependency via a structured parameterization. Specifically, we parameterize $y^\star(x, \xi)$ as $y_{\Phi^{[N]}}(W(x), \xi) \triangleq W(x)\Phi^{[N]}(\xi)$, where $W(x) \in \mathbb{R}^{d_y \times N}$ is a context-independent coefficient matrix, and $\Phi^{[N]} : \Xi \to \mathbb{R}^N$ is a vector of basis functions that captures the nonlinear dependence on $\xi$. We then consider the following SBO problem:

$$\min_{x \in \mathbb{R}^{d_x}} \quad F_{\Phi^{[N]}}(x) \triangleq \mathbb{E}_{(\xi,\eta)\sim\mathbb{P}_{(\xi,\eta)}}\left[f_{\Phi^{[N]}}(x, W^\star(x), \xi, \eta)\right] \qquad\qquad (\text{SBO}_{\Phi^{[N]}})$$

$$\text{s.t.} \quad W^\star(x) = \underset{W \in \mathbb{R}^{d_y \times N}}{\arg\min} \ \mathbb{E}_{(\xi,\eta)\sim\mathbb{P}_{(\xi,\eta)}}\left[g_{\Phi^{[N]}}(x, W, \xi, \eta)\right], \quad \forall x \in \mathbb{R}^{d_x}$$

where $f_{\Phi^{[N]}}(x, W, \xi, \eta) \triangleq f(x, y_{\Phi^{[N]}}(W, \xi), \xi, \eta)$ and $g_{\Phi^{[N]}}(x, W, \xi, \eta) \triangleq g(x, y_{\Phi^{[N]}}(W, \xi), \xi, \eta)$. Importantly, the lower-level solution is function of $x$ only, and the linearity of the parameterization $y_{\Phi^{[N]}}$ in $W$ preserves strong convexity of the lower-level problem, provided that $\Phi$ is not

| | Oracle | SBO with $m$ subproblems | CSBO | CSBO + conditions of Theorem 4.5 |
|---|---|---|---|---|
| Guo et al. (2021) | Conditional | $O\left(m\epsilon^{-3}\right)$ | - | - |
| Hu et al. (2023b) | Conditional | $\tilde{O}\left(\epsilon^{-4}\right)$ | $\tilde{O}\left(\epsilon^{-4}\right)$ | $\tilde{O}\left(\epsilon^{-4}\right)$ |
| Ours | Joint only | $\tilde{O}\left(\epsilon^{-3}\right)$ | $\tilde{O}\left(\epsilon^{-3}\operatorname{poly}\left(M_\Phi(\epsilon), \frac{1}{m_\Phi(\epsilon)}\right)\right)$ | $\tilde{O}\left(\epsilon^{-3}\right)$ |

Table 1: Comparison of sampling oracle assumptions and sample complexity bounds for SBO/CSBO.

ill-conditioned. Crucially, the lower-level objective is now expressed as an expectation over the joint distribution $\mathbb{P}_{(\xi,\eta)}$, eliminating the need for conditional sampling.

Our framework begins by selecting a vector of basis functions of $\Phi^{[N]}$ to reformulate (CSBO) as a standard SBO instance (SBO$_{\Phi^{[N]}}$). We then apply existing SBO algorithms to compute an $\epsilon$-accurate solution to (SBO$_{\Phi^{[N]}}$), which we lift to a corresponding solution to (CSBO) that is $\epsilon$-accurate if $\Phi^{[N]}$ is suitably expressive. Our analysis relates the regularities of (SBO$_{\Phi^{[N]}}$) and the optimality gap of the constructed solution to simple metrics on $\Phi^{[N]}$. These relationships not only guide the choice of basis functions but also enable near-optimal sample complexity guarantees under appropriate conditions on $\Phi^{[N]}$.

MAIN CONTRIBUTIONS

- We propose a reduction framework for CSBO that leverages the efficiency of existing SBO algorithms. Importantly, our approach requires only samples from the joint distribution $\mathbb{P}_{(\xi,\eta)}$, eliminating the need for conditional sampling from $\mathbb{P}_{\eta|\xi}$. We show that when a basis $\Phi$ is sufficiently expressive, the constructed solution is an $\epsilon$-stationary point to (CSBO) (Theorem 4.3).

- We relate the regularity constants of the reformulated SBO problem (SBO$_{\Phi^{[N]}}$), and its sample complexity, to simple metrics on $\Phi$ (Theorem 4.4). Crucially, this result establishes sufficient conditions for a basis to guarantee $\epsilon$-stationary solutions to (CSBO) with $\tilde{\mathcal{O}}(\epsilon^{-3})$ iterations and sample complexity, improving the best-known complexity for CSBO by an order of magnitude and matching the lower bound for nonconvex stochastic optimization up to logarithmic factors. We then show that Chebyshev polynomials are a practical, concrete choice of basis that achieves $\tilde{\mathcal{O}}(\epsilon^{-3})$ sample complexity under mild conditions on $g$ and $\mathbb{P}_\xi$ satisfied in many relevant settings (Theorem 4.5). Table 1 compares prior SBO and CSBO methods with our approach in terms of sampling oracle requirements and sample complexity.

- We validate our framework empirically on inverse optimization and hyperparameter tuning tasks. Our method consistently outperforms baselines in both convergence rate and final loss. It also exhibits greater flexibility in allocating resources to challenging context regions, matching or surpassing other methods with fewer samples and reduced memory usage.

## 2 RELATED WORK

Our work lies at the intersection of three lines of work: Stochastic Bilevel Optimization, Contextual Optimization, and Approximation Theory.

**Stochastic Bilevel Optimization**

Stochastic bilevel optimization (SBO), first introduced by Bracken & McGill (1973), has been extensively studied with a focus on the convergence and sample complexity of stochastic gradient methods (Ghadimi & Wang, 2018; Chen et al., 2021; Khanduri et al., 2021; Hong et al., 2023; Kwon et al., 2024; Chen et al., 2024). These works primarily address the standard setting in which the lower-level problem is independent of any external context.

The CSBO framework of Hu et al. (2023b) extends SBO by allowing the lower-level problem to vary with a context $\xi$, which introduces significant algorithmic and theoretical challenges. Their method leverages multilevel Monte Carlo estimators to speed up lower-level optimization, but a new lower-level problem is solved without warm start for each sample $\xi$. In contrast, our work introduces a reduction-based framework that transforms any CSBO instance into a structured SBO instance by parameterizing the lower-level solution across contexts. This enables the direct application of state-of-the-art SBO solvers – including first-order and variance-reduced methods (Guo et al., 2021; Khanduri et al., 2021; Dagréou et al., 2022; Chen et al., 2023; Shen & Chen, 2023; Kwon et al., 2023; Lu & Mei, 2024) – without requiring conditional sampling or prior knowledge on $y^\star$. Moreover, we prove that our approach achieves near-optimal sample complexity under mild assumptions, thereby closing the computational gap between SBO and CSBO for a broad class of problems.

**Contextual Optimization**

Contextual Optimization (CO) refers to the problem of selecting an action $z^\star(x)$ that minimizes the expected cost conditioned on a context or covariate $x$. A recent survey by Sadana et al. (2024) provides a comprehensive overview of this literature. CO is closely related to the lower-level problem in CSBO, as both involve making context-dependent decisions under uncertainty.

Common approaches to CO include optimizing surrogate losses aligned with downstream decision objectives (Elmachtoub & Grigas, 2022; Bennouna et al., 2024), or learning decision rules such as linear (Ban & Rudin, 2019) or neural network-based policies (Oroojlooyjadid et al., 2020). Our parameterization can be interpreted as a structured linear policy in the context variable $\xi$, where expressiveness is achieved via a nonlinear basis transformation $\Phi(\xi)$. Unlike most CO formulations, however, our decision rule arises as the solution to a bilevel problem where the upper-level objective depends on the learned representation of the lower-level solution. This structural coupling distinguishes our setting and motivates our complexity and approximation analysis.

**Approximation Theory**

Function approximation techniques have been employed to address SBO and other nested optimization problems. Recent work by Petrulionyte et al. (2024) directly optimizes over function spaces in SBO. More commonly, neural network approximations have been used to solve Lagrangian relaxations (Lv et al., 2008) or to learn surrogates for the upper- or lower-level objective (Patel et al., 2022; Kronqvist et al., 2023; Dumouchelle et al., 2023; 2024). While these models offer high expressiveness, their nonlinearity and overparameterization make it difficult to obtain tight or interpretable error guarantees.

In contrast, our approach leverages classical tools from approximation theory, particularly orthogonal polynomials. Fourier and Chebyshev polynomials are well-known for their strong convergence properties (Boyd, 2001; Gautschi, 2004; Chihara, 2011). In particular, the uniform error of approximation of Chebyshev polynomials decreases exponentially when approximating analytic functions, and algebraic convergence for smooth functions (Trefethen, 2019). We use this property to construct expressive and well-conditioned approximations to the context-dependent solution $y^\star(x, \cdot)$, and we derive explicit error bounds linking the basis approximation quality to the optimality gap of the original CSBO problem.

## 3 PRELIMINARIES

We first introduce the notation adopted throughout the paper. We use $\|\cdot\|$ to denote the $l_2$ norm of a vector and the spectral norm of a matrix. The variance of random vector or matrix $X$ is defined as $\mathbb{E}\left[\|X - \mathbb{E}[X]\|^2\right]$. For brevity of notation, we refer to $\{0, 1, ..., n\}$ as $[n]$. We denote by $\mathbb{E}_\xi$ and $\text{Var}_\xi$ the operators $\mathbb{E}_{\xi \sim \mathbb{P}_\xi}$ and $\text{Var}_{\xi \sim \mathbb{P}_\xi}$. Similarly, we denote by $\mathbb{E}_{\eta|\xi}$ and $\text{Var}_{\eta|\xi}$ the operators $\mathbb{E}_{\eta \sim \mathbb{P}_{\eta|\xi}}$ and $\text{Var}_{\eta \sim \mathbb{P}_{\eta|\xi}}$.

Throughout the paper, we make the following assumptions about problem (CSBO).

**Assumption 3.1.**

(i) The functions $f$, $\nabla f$, $\nabla g$, and $\nabla^2 g$, are $L_{f,0}$, $L_{f,1}$, $L_{g,1}$, and $L_{g,2}$-Lipschitz continuous with respect to $(x, y)$ for any fixed $(\xi, \eta)$, respectively.

(ii) The function $g$ is $\mu$-strongly convex with respect to $y$ for all $(x, \xi, \eta)$.

(iii) If $\eta \sim \mathbb{P}_{\eta|\xi}$, then the gradients $\nabla f(x, y, \xi, \eta)$, $\nabla g(x, y, \xi, \eta)$, and $\nabla^2 g(x, y, \xi, \eta)$, are unbiased and have variance bounded by $\sigma_f^2$, $\sigma_{g,1}^2$, and $\sigma_{g,2}^2$, respectively, uniformly across all $x$, $y$, and $\xi$.

*Remark* 3.2. Assumptions 3.1-(i) and 3.1-(ii) are standard in the SBO literature (Ghadimi & Wang, 2018; Yang et al., 2021; Hong et al., 2023). Assumption 3.1-(iii) is also common in CSBO (Hu et al., 2023b; Thoma et al., 2024). However, in contrast to (Hu et al., 2023b), we do not enforce the global Lipschitz continuity of $g$, thereby avoiding the implicit requirement that the lower-level variable $y$ lie in a bounded set. Moreover, we only require the variance of $\nabla f(x, y, \xi, \eta)$ to be bounded with respect to the conditional distribution $\mathbb{P}_{\eta|\xi}$, instead of the joint distribution $\mathbb{P}_{(\xi,\eta)}$.

We formally define the terms *feature map* and *expressive basis* which will be used frequently.

**Definition 3.3.** A *feature map* $\Phi^{[N]} : \Xi \to \mathbb{R}^N$ is a mapping whose components are the first $N$ elements of a countable basis of functions $\Phi$.

**Definition 3.4.** A countable basis of $\mathcal{F}(\Xi, \mathbb{R})$, $\Phi = \{\varphi_k\}_{k\in\mathbb{N}}$, is $N_\Phi$-*expressive* if the function $N_\Phi : \mathbb{R}_+^* \to \mathbb{N}^*$ is such that, for any $\epsilon > 0$, there exists $W^\dagger : \mathbb{R}^{d_x} \to \mathbb{R}^{d_y \times N_\Phi(\epsilon)}$ satisfying:

$$\mathbb{E}_\xi \left\| y_{\Phi^{[N_\Phi(\epsilon)]}}(W^\dagger(x), \xi) - y^\star(x, \xi) \right\|^2 \leq \frac{\epsilon^2}{4K^2} \frac{\mu}{L_{g,1}}, \quad \forall x \in \mathbb{R}^{d_x}, \tag{1}$$

where $K = L_{f,1} + \frac{L_{g,2} L_{f,0}}{\mu} + \frac{L_{g,2} L_{g,1} L_{f,0}}{\mu^2} + \frac{L_{f,1} L_{g,1}}{\mu}$ is a constant depending only on the regularities of $f$ and $g$. In that case, we refer to $\Phi^{[N_\Phi(\epsilon)]}$ as $\Phi^\epsilon$ for conciseness.

*Example* 3.5. Consider a CSBO problem where $y^\star(x, \xi) = \cos(\xi \cdot \cos(x))$ and $\mathbb{P}_\xi = U(0, 1)$. Let $\Phi$ be the polynomial basis $\Phi = (1, \xi, \xi^2, ...)$. Then $\Phi^{[N]} \triangleq (1, ..., \xi^{N-1})$ and the Taylor series of $y^\star$ at $\xi = 0$ gives for $N$ even and $\xi \in [0, 1]$: $\left| y^\star(x, \xi) - W(x)\Phi^{[N]}(\xi) \right| \leq \frac{|\cos(x)\xi|^N}{N!} \leq \frac{1}{N!}$. Thus $\Phi$ is $N_\Phi(\epsilon)$-*expressive* with $N_\Phi(\epsilon) = O(\ln \epsilon^{-1})$, and $\Phi^\epsilon \triangleq \Phi^{[N_\Phi(\epsilon)]} = (1, \xi, ..., \xi^{N_\Phi(\epsilon)-1})$.

# 4 REDUCTION FRAMEWORK AND SAMPLE COMPLEXITY GUARANTEES

Central to our approach is the parameterization of the lower-level solution using an expressive basis $\Phi$ to reformulate (CSBO) into (SBO$_{\Phi^\epsilon}$), an SBO problem with a single lower level. We address two main questions: (i) can an $\epsilon$-accurate solution to (CSBO) be constructed from a solution to (SBO$_{\Phi^\epsilon}$)? (ii) does the reformulated problem preserve the structural assumptions (e.g., strong convexity, Lipschitz continuity) required for applying efficient SBO algorithms? In this section, we answer both questions affirmatively and show how this enables efficient solutions to (CSBO).

## 4.1 HYPERGRADIENT APPROXIMATION AND REDUCTION VALIDITY

We begin by bounding the discrepancy between the hypergradients of (CSBO) and (SBO$_{\Phi^\epsilon}$).

**Proposition 4.1.** *Under assumption 3.1, the following holds for any $x \in \mathbb{R}^{d_x}$:*
$$\|\nabla F(x) - \nabla F_{\Phi^\epsilon}(x)\| \leq K \cdot \mathbb{E}_\xi \|y_{\Phi^\epsilon}(W^\star(x), \xi) - y^\star(x, \xi)\| \tag{2}$$
*where $K$ is defined in Definition 3.4.*

This result shows that if the parameterized solution $y_{\Phi^\epsilon}(W^\star(x), \xi)$ closely approximates $y^\star(x, \xi)$ in expectation, then the hypergradient of the surrogate objective $F_{\Phi^\epsilon}$ is a good approximation to that of the true objective $F$. However, directly bounding the right-hand side of equation 2 is nontrivial, as $W^\star(x)$ is characterized implicitly. The regularities of $g$ and the optimality of $W^\star(x)$ in (SBO$_{\Phi^\epsilon}$) yield:

**Proposition 4.2.** *Under assumption 3.1, we have for any $x \in \mathbb{R}^{d_x}$ and $W \in \mathbb{R}^{d_y \times N_\Phi(\epsilon)}$:*
$$\mathbb{E}_\xi \|y_{\Phi^\epsilon}(W^\star(x), \xi) - y^\star(x, \xi)\|^2 \leq \frac{2L_{g,1}}{\mu} \cdot \mathbb{E}_\xi \|y_{\Phi^\epsilon}(W, \xi) - y^\star(x, \xi)\|^2. \tag{3}$$

This result allows us to control the right-hand side of equation 2 by evaluating the error at a suitable $W$ for which the expected distance between $y_{\Phi^\epsilon}(W, \cdot)$ and $y^\star(x, \cdot)$ is well characterized. Since expressive bases admit such $W$ for any fixed $\epsilon$, it is sufficient to solve the surrogate problem (SBO$_{\Phi^\epsilon}$) to produce an $\epsilon$-stationary solution to the original problem (CSBO).

**Theorem 4.3.** *Suppose that assumption 3.1 holds, and let $\Phi$ be an expressive basis. If $(x^\star, W^\star(x^\star))$ is an $\frac{\epsilon}{\sqrt{2}}$-stationary solution to (SBO$_{\Phi^\epsilon}$), then $(x^\star, \xi \mapsto y_{\Phi^\epsilon}(W^\star(x^\star), \xi))$ is an $\epsilon$-stationary solution to (CSBO).*

*Proof.* Combining Propositions 4.1 and 4.2, we obtain:

$$
\begin{aligned}
\mathbb{E}\|\nabla F(x^\star)\|^2 &\leq \mathbb{E}\|\nabla F_{\Phi^\epsilon}(x^\star)\|^2 + \mathbb{E}\|\nabla F(x^\star) - \nabla F_{\Phi^\epsilon}(x^\star)\|^2 \\
&\overset{(1)}{\leq} \frac{\epsilon^2}{2} + \mathbb{E}\left[ K^2 \left( \mathbb{E}_\xi \|y_{\Phi^\epsilon}(W^\star(x^\star), \xi) - y^\star(x^\star, \xi)\| \right)^2 \right] \\
&\overset{(2)}{\leq} \frac{\epsilon^2}{2} + \mathbb{E}\left[ \frac{2K^2 L_{g,1}}{\mu} \mathbb{E}_\xi \|y_{\Phi^\epsilon}(W^\dagger(x^\star), \xi) - y^\star(x^\star, \xi)\|^2 \right] \\
&\overset{(3)}{\leq} \frac{\epsilon^2}{2} + \frac{2K^2 L_{g,1}}{\mu} \cdot \frac{\epsilon^2 \mu}{4K^2 L_{g,1}} \\
&\leq \epsilon^2
\end{aligned}
$$

where (1) uses Proposition 4.1 and the fact that $(x^\star, W^\star(x^\star))$ is an $\frac{\epsilon}{\sqrt{2}}$-stationary solution to (SBO$_{\Phi^\epsilon}$), (2) results from Proposition 4.2 and Jensen's inequality, and (3) holds since $\Phi^\epsilon$ satisfies equation 1. □

This theorem essentially states that by working in the $\Phi^\epsilon$-parametrized space, any off-the-shelf SBO solver can be used to solve the CSBO instance. Since the upper and lower level expectations of (SBO$_{\Phi^\epsilon}$) are both taken over $\mathbb{P}_{(\xi,\eta)}$, one can obtain a solution to CSBO without conditional sampling oracle. Moreover, if $\Phi$ induces a smooth map $\xi \mapsto y_{\Phi^\epsilon}(x, \xi)$, then a gradient step at a given $\xi$ generalizes to its neighbors, yielding computational efficiency and guarding against overfitting to context-specific noise.

## 4.2 GENERAL SAMPLE COMPLEXITY

Although expressive bases guarantee a small approximation error, achieving the condition in equation 1 may require $\Phi^\epsilon$ to have a large cardinality. In particular, the lower-level problem in (SBO$_{\Phi^\epsilon}$) may become high-dimensional or poorly conditioned as $\epsilon \to 0$, especially if the basis functions are unbounded or highly correlated. To quantify these effects, we define the following metrics:

$$
M_\Phi(\epsilon) = \max\left( 1, \sup_{\xi \in \Xi} \|\Phi^\epsilon(\xi)\| \right), \quad \text{and} \quad m_\Phi(\epsilon) = \min\left( 1, \lambda_{\min}(\Sigma_{\Phi^\epsilon}) \right)
$$

where $\Sigma_\Phi \triangleq \mathbb{E}_\xi \left[ \Phi^\epsilon(\xi) \Phi^\epsilon(\xi)^\top \right]$ is the covariance matrix of the feature map $\Phi^\epsilon$. Here, $M_\Phi(\epsilon)$ upper bounds the maximum magnitude attainable by the feature map $\Phi^\epsilon$, whereas $m_\Phi(\epsilon)$ captures the non-degeneracy of $\Phi^\epsilon$ by lower bounding the smallest eigenvalue of $\Sigma_{\Phi^\epsilon}$. A small $m_\Phi(\epsilon)$ indicates that the feature vectors are concentrated near a lower-dimensional subspace. The reformulation's lower level inherits strong convexity if and only if $m_\Phi(\epsilon)$ is positive. To ensure that the lower level of the reformulation remains strongly-convex as $\epsilon$ tends to 0 for our asymptotic complexity analysis, we restrict our attention to bases satisfying $m_\Phi(\epsilon) > 0$ for all $\epsilon > 0$ and refer to any such basis as *well-conditioned*.

We show that the reformulation (SBO$_{\Phi^\epsilon}$) can be solved efficiently by verifying that the functions $f_{\Phi^\epsilon}(x, W, \xi, \eta)$, $g_{\Phi^\epsilon}(x, W, \xi, \eta)$ and their first and second order derivatives satisfy the assumptions of Guo et al. (2021) provided in the Appendix [D.1, D.2]. Importantly, we express in lemmas C.4 and C.7 the Lipschitz constants and variance bounds in terms of $M_\Phi(\epsilon)$ and $m_\Phi(\epsilon)$. Substituting these regularity coefficients into Theorem 1 of Guo et al. (2021), we obtain the result below.

**Theorem 4.4.** *Suppose that assumption 3.1 holds and that $\Phi$ is well-conditioned. Then, an $\epsilon$-stationary solution to (SBO$_{\Phi^\epsilon}$) can be achieved with a sample complexity in $\tilde{\mathcal{O}}\left( \frac{1}{\epsilon^3} \cdot poly\left( M_\Phi(\epsilon), \frac{1}{m_\Phi(\epsilon)} \right) \right)$.*

Fundamentally, this Theorem shows that the sample complexity of (SBO$_{\Phi^\epsilon}$) scales polynomially in $M_\Phi(\epsilon)$ and $1/m_\Phi(\epsilon)$. While $N_\Phi(\epsilon)$ is a natural proxy for lower-level complexity (and hence for

sample complexity), Theorem 4.4 shows that the complexity is governed by subtler interactions between $\Phi$ and $\mathbb{P}_\xi$ captured by $M_\Phi(\epsilon)$ and $m_\Phi(\epsilon)$. Specifically, the complexity remains well-controlled when:

1. The magnitude of $\Phi^\epsilon$ grows slowly, ensuring that the Lipschitz and smoothness constants of the reformulation remain comparable to those of the original problem, and avoiding the need for significantly smaller step-sizes.

2. The conditioning of $\Phi^\epsilon$ decreases slowly, guaranteeing that the reformulation's lower level retains a pronounced strong convexity.

Under these two favorable regimes, the number of iterations and samples required to obtain an $\epsilon$-accurate solution to (SBO$_{\Phi^\epsilon}$) is tightly controlled. Crucially, any expressive basis $\Phi$ satisfying $M_\Phi(\epsilon) = \tilde{\mathcal{O}}(1)$ and $1/m_\Phi(\epsilon) = \tilde{\mathcal{O}}(1)$ results in a near-optimal sample complexity $\tilde{\mathcal{O}}(\epsilon^{-3})$.

In example 3.5, the monomial basis yields $N_\Phi(\epsilon) = O(\ln(\epsilon^{-1}))$ and $M_\Phi(\epsilon) = \sqrt{N_\Phi(\epsilon)}$. Moreover, the covariance matrix $\Sigma_{\Phi^\epsilon}$ coincides with the Hilbert matrix, implying $m_\Phi(\epsilon) = \Theta\left(\sqrt{N_\Phi(\epsilon)}\left(1 + \sqrt{2}\right)^{-4N_\Phi(\epsilon)}\right)$. Thus, under the uniform distribution, the monomial basis becomes rapidly ill-conditioned as $\epsilon \to 0$, despite the fact that $N_\Phi(\epsilon)$ grows only logarithmically. In contrast, the Chebyshev basis satisfies $N_{\text{Chebyshev}}(\epsilon) = O(\ln(\epsilon^{-1}))$ and $M_{\text{Chebyshev}}(\epsilon) \leq \sqrt{N_{\text{Chebyshev}}(\epsilon)}$, while remaining well-conditioned with $1/m_{\text{Chebyshev}}(\epsilon) = O(N_{\text{Chebyshev}}(\epsilon))$ for any $\epsilon > 0$. Theorem 4.4 thus yields a sample complexity in $\tilde{\mathcal{O}}(\epsilon^{-3})$ for this specific choice of basis.

### 4.3 CHEBYSHEV BASIS AND NEAR-OPTIMAL SAMPLE COMPLEXITY

Theorems 4.3 and 4.4 established that any sufficiently expressive, well-conditioned basis yields an $\epsilon$-stationary solution to CSBO problems with a sample complexity depending on simple metrics involving the basis and context distribution. Example 3.5 illustrates that the basis choice is key to obtain strong convergence guarantees. We now provide sufficient conditions, satisfied in many practical settings, under which the basis constructed from Chebyshev polynomials (defined in E.1) achieves near-optimal complexity guarantees. We emphasize that these conditions are not necessary, and that Chebyshev polynomials and other bases can yield near-optimal complexity when these conditions are violated.

**Theorem 4.5.** *Suppose that Assumption 3.1 holds, along with the following conditions:*

*(c.1) Either (i) there exists $\underline{c} > 0$ such that the density of $\mathbb{P}_\xi$ is lower bounded by $\underline{c}$ on $\Xi$, or (ii) the cardinality of $\Xi$ is finite.*

*(c.2) The support $\Xi$ is bounded.*

*(c.3) The function $G(x, y, \xi) \triangleq \mathbb{E}_{\eta|\xi}\left[g(x, y, \xi, \eta)\right]$ is analytic in $(y, \xi)$ for any fixed $x$.*

*Then the Chebyshev polynomial basis $\Phi$ is well-conditioned with:*

$$M_\Phi(\epsilon) = \tilde{\mathcal{O}}\left(1\right), \quad \text{and} \quad m_\Phi^{-1}(\epsilon) = \tilde{\mathcal{O}}\left(1\right).$$

The conditions of Theorem 4.5 ensure that $y^\star(x, \cdot)$ can be efficiently approximated by a truncated Chebyshev series. Specifically, (c.1) prevents the feature covariance matrix $\Sigma_{\Phi_\epsilon}$ from becoming degenerate as $\epsilon \to 0$; (c.2) avoids requiring periodicity or decay conditions at infinity for uniform approximation; and (c.3) guarantees analyticity of $y^\star(x, \cdot)$, so that the uniform error of approximation of Chebyshev series decreases exponentially fast (Bernstein, 1912; Adcock & Huybrechs, 2014; Trefethen, 2019). Many applications satisfy these conditions, including Meta-Learning and Wasserstein DRO with Side Information (Hu et al., 2023b), as well as Tax Design, Reward Shaping, and Dynamic Mechanism Design (Thoma et al., 2024), which are naturally modeled with discrete or truncated continuous context distributions.

The proof builds on this exponentially fast decay for functions that are analytic in an open region bounded by a Bernstein ellipse, and extends these convergence results to multivariate analytic functions. As real-analytic functions on a bounded set $\Xi$ are analytically continuable in an open complex set containing $\Xi$, we use our results on multivariate analytic functions to prove that the

Chebyshev basis is expressive with $N_\Phi$ growing sub-polynomially in $\epsilon^{-1}$. Together with the facts that Chebyshev polynomials are uniformly bounded by 1, and that the degeneracy of the truncated series is bounded as $m_\Phi(\epsilon) = \Omega\left(1/N_\Phi(\epsilon)\right)$ under condition (c.1), we obtain the desired result.

Along with theorems 4.3 and 4.4, it follows that one can reduce (CSBO) to (SBO$_{\Phi\epsilon}$) using the Chebyshev basis and solve the latter with $\tilde{\mathcal{O}}(\epsilon^{-3})$ samples as the terms involving $M_\Phi(\epsilon)$ and $1/m_\Phi(\epsilon)$ in theorem 4.4 collapse into polylogarithmic factors.

**Corollary 4.6.** *Suppose that assumption 3.1 and the conditions of Theorem 4.5 hold. Then, an $\epsilon$-stationary solution to (CSBO) can be achieved with a near-optimal sample complexity of $\tilde{\mathcal{O}}\left(\epsilon^{-3}\right)$.*

In practice, the conditions of Theorem 4.5 are satisfied in many relevant settings where $\xi$ is finite (e.g. SBO with multiple subproblems) or has bounded support and non-vanishing density (e.g., uniform, truncated Gaussian, or mixture distributions). Notably, meta-learning and personalized federated learning, where $\xi$ ranges over a finite number of tasks, and robust stochastic optimization with side information drawn from a distribution with full support on a compact set all naturally satisfy these conditions.

## 5 NUMERICAL RESULTS

Many machine learning tasks can be formulated as CSBO, including hyper-parameter optimization (Shaban et al., 2019; Franceschi et al., 2018), inverse optimization (Ahuja & Orlin, 2001), meta-learning (Rajeswaran et al., 2019), reinforcement learning from human feedback (Chakraborty et al., 2024), and personalized federated learning (Shamsian et al., 2021).

In the following experiments, we compare our proposed reduction framework against STOCBIO (Ji et al., 2021) applied to discretized SBO approximations of the original CSBO problems. For STOCBIO[$N$], the context space $\Xi$ is partitioned into $N$ intervals of equal size, each treated as an independent subproblem, yielding a SBO formulation with $N$ subproblems.

Our method, denoted $\mathcal{R}_\Phi[N]$, uses the feature map $\Phi^{[N]}$ and STOCBIO[1] to solve the reduced problem. We experiment with monomial, Chebyshev, and Fourier bases; however, in the hyperparameter optimization task, we report only Chebyshev and monomial results, as Fourier and Chebyshev perform nearly identically in that setting. Importantly, for any fixed $N$, STOCBIO[$N$] and $\mathcal{R}_\Phi[N]$ have identical memory usage: $d_x + N d_y$, enabling a fair comparison under fixed resource constraints. Details of the numerical experiments are provided in Appendix A.

### 5.1 INVERSE OPTIMIZATION

Inverse optimization seeks parameters $x$ that explain observed decisions $\eta$ as optimal solutions to an underlying optimization problem. While traditionally approached through bilevel formulations, many inverse problems naturally fall into the CSBO setting, yet are rarely treated as such in existing work.

We consider the Static Traffic Assignment (STA) problem (Beckmann et al., 1956), which models network equilibrium flows under fixed origin-destination (OD) demand. Using Beckmann's convex-potential formulation and introducing a penalty for constraint violations, we cast inverse capacity estimation as the following CSBO problem:

$$\min_x \quad \mathbb{E}_{(\xi,\eta)\sim\mathbb{P}_{(\xi,\eta)}} \|y^\star(x,\xi) - \eta\| \tag{4}$$

$$\text{s.t.} \quad y^\star(x,\xi) = \arg\min_y \sum_{e\in\mathcal{E}} \int_0^{y_e} t_e(z;x)dz + \lambda_\xi(y)$$

where $y^\star(x,\xi)$ is the equilibrium flow given edge capacities $x$ and OD demand $\xi$, $\eta$ is the corresponding noisy observation of that flow, $t_e$ is the edge performance function relating the flow to the travel time, and $\lambda_\xi(y)$ penalizes infeasible flows and the violation of OD pairs demand constraints.

We simulate a two-edge network with a single OD pair. For every trial, we sample a ground truth capacity vector $x^\star$ and generate training and test sets with $10^3$ i.i.d. samples $(\xi,\eta)$ each. The final solution $(\bar{x},\bar{y}(\cdot))$ is averaged over the last $10\%$ epochs. Performance is evaluated via the test loss $F(\bar{x})$, evaluated over the test set after computing the exact lower-level solutions $y^\star(\bar{x},\xi)$. We also

report the expected lower-level error $\Delta_y \triangleq \mathbb{E}_\xi \|y^\star(\bar{x}, \xi) - \bar{y}(\xi)\|^2$ and the upper-level parameter error $\Delta_x \triangleq \|\bar{x} - x^\star\|^2$.

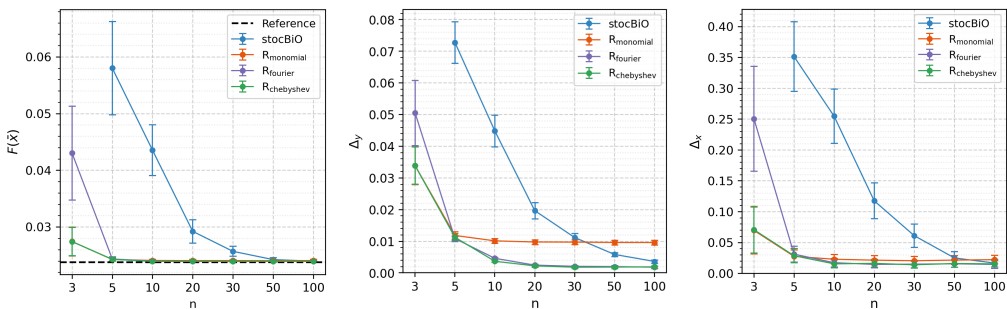

Figure 1: Loss $F(\bar{x})$ (left), lower level solution error $\Delta_y$ (center), and upper level solution error $\Delta_x$ (right) of STOCBIO and our reduction framework using monomial, Fourier, and Chebyshev bases. The reference loss is $F_{\text{ref}} = F(x^\star)$.

Figure 1 demonstrates the advantage of the CSBO reduction. As few as $N = 3$ basis functions suffice to produce reasonable solutions, whereas STOCBIO struggles with instability or excessive sample requirements. By $N = 5$, all three bases close the optimality gap to within $3\%$, while STOCBIO achieves a comparable accuracy at $N = 50$. Beyond $N = 5$, the lower-level error stops decreasing significantly for the monomial basis, while the more expressive Chebyshev and Fourier bases continue to reduce the error and consistently outperform STOCBIO at every $N$.

$\mathcal{R}_{\text{Chebyshev}}$ and $\mathcal{R}_{\text{Fourier}}$ achieve the smallest loss for $N \geq 10$, whereas STOCBIO requires $N \geq 100$, and still does not reach the same lower-level error. These results validate our theoretical claims: expressive bases can compactly approximate context-sensitive solutions in CSBO. Compared to partition-based SBO approximations, the parameterization leverages the continuity of $y^\star$ to achieve better accuracy using an order of magnitude less memory or, with the same memory budget, an order of magnitude lower optimality gap.

## 5.2 Hyperparameter Optimization

Hyperparameter optimization has become the benchmark of choice for evaluating the performance of stochastic bilevel optimization (SBO) methods in practical, high-dimensional settings. Hu et al. (2023a) introduced a more challenging variant of this problem involving temperature-dependent objectives. Instead of simplifying the problem to a finite set of temperatures, we address the original formulation in which temperatures are drawn from a continuous distribution.

In this experiment, we aim to train a linear classifier for MNIST (LeCun et al., 1998) with the complication that a fixed fraction $p = 0.3$ of the training labels have been randomly corrupted. Here, the upper level seeks to weight the training points to minimize the expected validation loss across a distribution of model temperatures $\mathbb{P}_\xi$, while the lower level computes, for each $\xi$, the model parameters $y$ by minimizing a weighted and regularized training loss. Formally:

$$\min_x \quad \mathbb{E}_{\xi \sim \mathbb{P}_\xi} \mathbb{E}_{D \sim \mathcal{D}_{\text{val}}} \left[ \mathcal{L}_\xi(y^\star(x, \xi); D) \right] \tag{5}$$

$$\text{s.t.} \quad y^\star(x, \xi) = \arg\min_y \mathbb{E}_{D \sim \mathcal{D}_{\text{train}}} \left[ \sigma(x) \mathcal{L}_\xi(y; D) + \lambda \|y\|^2 \right]$$

where $\mathcal{L}_\xi$ is a temperature-specific convex loss, and $\mathbb{P}_\xi$, $\mathcal{D}_{\text{val}}$, and $\mathcal{D}_{\text{train}}$, denote the distributions over temperatures, validation data, and training data, respectively. This CSBO formulation encourages robustness to label noise while generalizing across temperatures.

Figure 2 compares our method $\mathcal{R}_\Phi[N]$ against STOCBIO$[N]$ baselines. Algorithms that solve a separate inner problem for each sample $\xi$ are omitted since they require a prohibitively large number of iterations to converge, even with the acceleration method of Hu et al. (2023b). By discretizing the context space into $N$ subproblems, STOCBIO$[N]$ yields lower final validation loss for larger $N$ at the cost of slower convergence, since finer partitions require nearby regions of $\Xi$ to be optimized independently and redundant. For small $N$, the high and increasing training loss of STOCBIO$[N]$

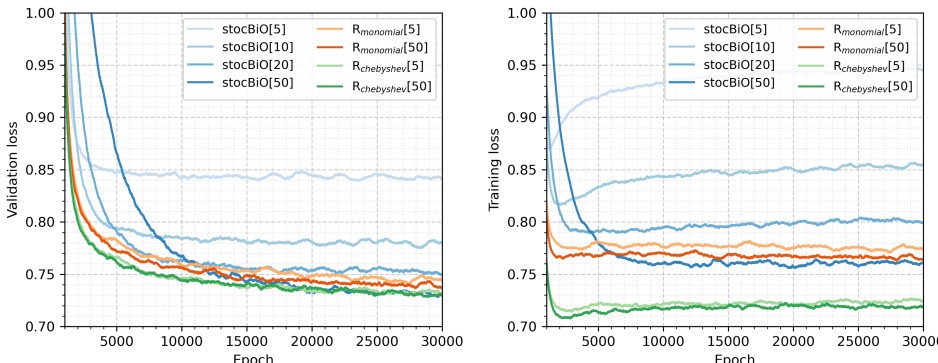

Figure 2: Moving average of the validation (left) and training (right) losses over epochs of stocBiO and monomial, Fourier, and Chebyshev bases on hyperparameter optimization.

(arising from the increase, on average, of the weights $\sigma(x)$) indicates its inability to accurately estimate $y^\star$ across all contexts and adapt to new training weights. Even for large $N$, it fails to accurately approximate $y^\star$ in regions where it varies sharply in $\xi$.

In contrast, our method leverages shared structure by learning global coefficients across all contexts, which cuts computational overhead and enables the model to allocate resources to context regions that require them most. As a result, $\mathcal{R}_{\text{Chebyshev}}[5]$ delivers both the fastest convergence and the lowest final training and validation losses; adding more basis functions yields only marginal gains. In comparison, the monomial basis lacks the expressiveness to substantially improve the accuracy of the lower-level solution approximation as $N$ grows, and thus performs worse than other methods for $N$ large.

## 6 CONCLUSION

We presented a framework that reduces any instance of CSBO to an SBO problem by parameterizing the lower-level solution with expressive basis functions. This reduction decouples the upper-level decision from the context and eliminates the need for conditional sampling oracles, which are often impractical in real-world applications. We related the sampling complexity to simple basis-dependent metrics, establishing criteria for achieving solutions with near-optimal $\tilde{\mathcal{O}}(\epsilon^{-3})$ complexity, an order of magnitude better than existing CSBO methods and matching the nonconvex lower bound up to logarithmic factors. We then showed that Chebyshev polynomials satisfy these criteria under mild conditions, offering a concrete and efficient choice of basis for a broad class of problems. We validated our method on inverse and hyperparameter optimization tasks, demonstrating faster convergence, lower final loss, and reduced memory usage compared to partition-based baselines. While we demonstrated that the Chebyshev basis is a versatile choice, online learning of instance-specific bases could yield a parameterization with improved accuracy and parsimony. Establishing practical algorithms and theoretical guarantees for adaptive bases is left for future work.

### ACKNOWLEDGMENTS

This work was supported by AFOSR Grant FA9550-23-1-0190. Additional support was provided to the author Jiawei Zhang by the Office of the Vice Chancellor for Research and Graduate Education at the University of Wisconsin-Madison with funding from the Wisconsin Alumni Research Foundation.

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

# A    SPECIFICATIONS AND FURTHER NUMERICAL EXPERIMENTS

Experiments are run in parallel on an AMD EPYC 9734 processor. Each individual run uses 16MB for a runtime ranging from 10 to 90 minutes, depending on the application and parameters used.

We omit the DL-SGD and RT-MLMC schemes Hu et al. (2023b) from our benchmarks because of their inner-loop strategy: both schemes use start from an arbitrary $y_0$ and use the EPOCH-GD algorithm of Hazan & Kale (2014) without the projection step. In practice, these schemes perform best when either:

1. $\{y^\star(x, \xi) : \xi \in \Xi\}$ is contained in a small ball around some $y_0(x)$, where $y_0(x)$ is known or estimable from previous iterations.

2. The lower-level problem is sufficiently well-conditioned that only a few inner-loop steps suffice for convergence.

In our experiments, such a $y_0(x)$ does not exist since the lower level solutions $y^\star(x, \cdot)$ vary substantially in $\xi$. Furthermore, depending on the inner-loop step size, the solution either diverges or requires a large number of steps to provide a reasonable estimate of $y^\star$ (and thus $\nabla F$), resulting in an excessive runtime.

## A.1    INVERSE OPTIMIZATION

In this experiment, we consider a network with two nodes, two edges, and a travel demand $\xi$ from node 0 to node 1. We use the edge performance function $t_e(y; x) \triangleq t_{0,e} \cdot \left( 1 + \alpha \left( \frac{y_e}{x_e} \right)^\beta \right)$ with $\alpha = 1$ and $\beta = 4$. To penalize negative flows and violation of the travel demand $\xi$, we define

$$\lambda_\xi(y) \triangleq \lambda_{\text{demand}} \cdot \left( \left( \xi - \sum_{e \in \mathcal{E}} y_e \right)^+ \right)^2 + \lambda_+ \cdot \sum_{e \in \mathcal{E}} \left( y_e^- \right)^2$$

where $z^+$ (resp. $z^-$) denotes the positive (resp. negative) part of $z$, $\lambda_{\text{demand}} = 100$, and $\lambda_+ = 50$. Note that one only needs to penalize insufficient total flow since the edge performance function is strictly increasing with respect to the flow $y$. The results are averaged over 50 runs, with 95% confidence intervals assuming normally distributed errors across runs. Each run uses a synthetic dataset generated with the following procedure. The free flow travel times $t_0$ and ground truth capacities $x^\star$ are sampled from the uniform distribution supported on $[1, 2]^{|\mathcal{E}|}$ and $[0.2, 0.8]^{|\mathcal{E}|}$, respectively. We generate $n_{\text{train}} = 1000$ training samples by drawing each context $\xi$ independently from a standard uniform distribution. For each $\xi$, we compute the corresponding optimal flow $y^\star(x^\star, \xi)$ using gradient descent, and set $\eta = y^\star(x^\star, \xi) + \epsilon$ where $\epsilon \sim \mathcal{N}(0, \sigma_0)$. We resample the noise $\epsilon$ until $\eta$ is non-negative. We follow the same procedure for the $n_{\text{test}} = 1000$ samples forming the test set. We follow the same steps as in the hyper-paramters optimization experiment to select $\alpha$, $\beta$, $T_{\text{inner}}$, and set $K = 10$ and $s = 10^{-3}$.

## A.2    HYPER-PARAMETERS OPTIMIZATION

We compare the performance of our proposed algorithm with stocBio, a reference algorithm that outperforms other baseline algorithms BSA, TTSA, HOAG on the MNIST Data Hyper-Cleaning task, a hyperparameter optimization problem. The dataset consists in 19,000 training and 1,000 validation images. The objective of Data hyper-cleaning involves training classifiers on a dataset where each label has been randomly and independently corrupted with probability $p$; that is, each label is replaced by a random class with chance $p$. The classifiers have losses with different temperatures. Formally, the objective function is:

$$\min_x \quad \mathbb{E}_{\xi \sim \mathbb{P}_\xi} \left[ \frac{1}{|\mathcal{D}_{\text{val}}|} \sum_{(X_i, Y_i) \in \mathcal{D}_{\text{val}}} L(y^\star(x, \xi) X_i / \xi), Y_i) \right]$$

$$\text{s.t.} \quad y^\star(x, \xi) = \arg\min_y \frac{1}{|\mathcal{D}_{\text{train}}|} \sum_{(X_i, Y_i) \in \mathcal{D}_{\text{train}}} \sigma(x) L(y X_i / \xi), Y_i) + \lambda \|y\|^2$$

where $L$ is the cross-entropy loss and $\sigma(\cdot)$ is the sigmoid function. Here $\mathbb{P}_\xi = \mathcal{U}(0.1, 10)$ and we choose the regularization parameter $\lambda = 10^{-3}$. The results are averaged over 20 randomized trials. We use a batch size of 512 and use grid search to choose: the inner-loop stepsize $\beta$ from $\{0.001, 0.01, 0.1, 1, 10\}$, the outer-loop stepsize $\alpha$ from $\{10^k \; : \; k \in [\![-5, 5]\!]\}$, and the number of inner-loop steps $T_{inner}$ from $\{1, 10, 100\}$. We choose a number of samples $K = 10$ and a scaling $s = 10^{-2}$ to approximate $\left(\mathbb{E}_{\eta \sim \mathbb{P}_{\eta|\xi}} \nabla_{22}^2 g(x, y, \xi, \eta)\right)^{-1}$ as $s \cdot \prod_{k=1}^{K} \left(I - s \cdot \nabla_{22}^2 g(x, y, \xi, \eta_n)\right)$.

## A.3 Additional SBO Backbones

To demonstrate that our findings are not tied to a particular choice of backbone solver, we include here two additional SBO solvers which differ substantially in design from STOCBIO: SUSTAIN Khanduri et al. (2021) and SOBA Dagréou et al. (2022). We follow the same setup as in the data-cleaning task. We compare the final training and validation losses of the solutions obtained on the $n$-subproblems SBO approximation (SOLVER) with the solutions obtained on the single lower-level reformulation using $n$ Chebyshev basis functions ($R_{\text{Chebyshev}}[\text{SOLVER}]$). The results are averaged over 5 randomly seeded runs. As shown in Table 2, the parametrization consistently improves both training and validation losses across all solvers for any fixed $n$. Further, a lower final loss can be achieved with fewer basis functions than subproblems. This indicates that the advantages of our parametrization are not specific to STOCBIO, but persist across several distinct SBO solvers.

Table 2: Performance comparison of various SBO solvers with and without reduction.

| Method | $n$ | Validation Loss | Training Loss |
|---|---|---|---|
| SOBA | 5 | $0.869 \pm 0.007$ | $1.208 \pm 0.021$ |
| | 10 | $1.015 \pm 0.011$ | $1.405 \pm 0.017$ |
| | 50 | $1.348 \pm 0.024$ | $1.838 \pm 0.034$ |
| $R_{\text{Chebyshev}}[\text{SOBA}]$ | 5 | $0.659 \pm 0.014$ | $0.790 \pm 0.030$ |
| | 10 | $0.653 \pm 0.013$ | $0.783 \pm 0.027$ |
| | 50 | $0.656 \pm 0.012$ | $0.786 \pm 0.029$ |
| SUSTAIN | 5 | $1.133 \pm 0.006$ | $1.335 \pm 0.025$ |
| | 10 | $1.167 \pm 0.010$ | $1.540 \pm 0.015$ |
| | 50 | $1.326 \pm 0.018$ | $1.887 \pm 0.018$ |
| $R_{\text{Chebyshev}}[\text{SUSTAIN}]$ | 5 | $0.890 \pm 0.011$ | $0.972 \pm 0.026$ |
| | 10 | $0.905 \pm 0.008$ | $0.975 \pm 0.012$ |
| | 50 | $0.903 \pm 0.009$ | $0.968 \pm 0.028$ |
| STOCBIO | 5 | $0.941 \pm 0.010$ | $0.848 \pm 0.027$ |
| | 10 | $0.837 \pm 0.020$ | $0.784 \pm 0.036$ |
| | 50 | $0.734 \pm 0.022$ | $0.739 \pm 0.033$ |
| $R_{\text{Chebyshev}}[\text{STOCBIO}]$ | 5 | $0.715 \pm 0.022$ | $0.740 \pm 0.041$ |
| | 10 | $0.706 \pm 0.010$ | $0.729 \pm 0.023$ |
| | 50 | $0.713 \pm 0.017$ | $0.733 \pm 0.026$ |

## B Similarity of Hypergradients

We first prove Proposition 4.1.

*Proof.* Let $x \in \mathbb{R}^{d_x}$. Recall that $\nabla F(x) = \mathbb{E}_{(\xi, \eta)}[\nabla F(x, \xi, \eta)]$ and $\nabla F_{\Phi^\epsilon}(x) = \mathbb{E}_{(\xi, \eta)}[\nabla F_{\Phi^\epsilon}(x, \xi, \eta)]$, where:

$$\nabla F(x, \xi, \eta) \triangleq \nabla_1 f(x, y^\star(x, \xi), \xi, \eta)$$
$$- \nabla_{12} g(x, y^\star(x, \xi), \xi, \eta) \left[\nabla_{22}^2 g(x, y^\star(x, \xi), \xi, \eta)\right]^{-1} \nabla_2 f(x, y^\star(x, \xi), \xi, \eta)$$
$$\nabla F_{\Phi^\epsilon}(x, \xi, \eta) \triangleq \nabla_1 f(x, y_\Phi(W^\star(x), \xi), \xi, \eta)$$
$$- \nabla_{12} g(x, y_\Phi(W^\star(x), \xi), \xi, \eta) \left[\nabla_{22}^2 g(x, y_\Phi(W^\star(x), \xi), \xi, \eta)\right]^{-1} \nabla_2 f(x, y_\Phi(W^\star(x), \xi), \xi, \eta)$$

Consider a sample $(\xi, \eta)$. For readability, we define $\Delta_y(x, \xi) \triangleq \|y_\Phi(W^\star(x), \xi) - y^\star(x, \xi)\|$ and:

$$A(y_\Phi) = \nabla_{12} g(x, y_\Phi(W^\star(x), \xi), \xi, \eta) \qquad A(y^\star) = \nabla_{12} g(x, y^\star(x, \xi), \xi, \eta)$$
$$B(y_\Phi) = \nabla_{22}^2 g(x, y_\Phi(W^\star(x), \xi), \xi, \eta) \qquad B(y^\star) = \nabla_{22}^2 g(x, y^\star(x, \xi), \xi, \eta)$$
$$C(y_\Phi) = \nabla_2 f(x, y_\Phi(W^\star(x), \xi), \xi, \eta) \qquad C(y^\star) = \nabla_2 f(x, y^\star(x, \xi), \xi, \eta)$$

We can then write and bound $\|\nabla F_{\Phi^\epsilon}(x, \xi, \eta) - \nabla F(x, \xi, \eta)\|$ using the triangular inequality as:

$$
\begin{aligned}
\|\nabla F_{\Phi^\epsilon}(x, \xi, \eta) - \nabla F(x, \xi, \eta)\| &\leq \|\nabla_1 f(x, y_\Phi(W^\star(x), \xi), \xi, \eta) - \nabla_1 f(x, y^\star(x, \xi), \xi, \eta)\| \\
&\quad + \|A(y_\Phi) B(y_\Phi)^{-1} C(y_\Phi) - A(y^\star) B(y^\star)^{-1} C(y^\star)\| \\
&\leq L_{f,1} \Delta_y(x, \xi) + \|A(y_\Phi) - A(y^\star)\| \cdot \|B(y^\star)^{-1}\| \cdot \|C(y^\star)\| \\
&\quad + \|B(y_\Phi)^{-1} - B(y^\star)^{-1}\| \cdot \|A(y_\Phi)\| \cdot \|C(y_\Phi)\| \\
&\quad + \|C(y_\Phi) - C(y^\star)\| \cdot \|A(y_\Phi)\| \cdot \|B(y^\star)^{-1}\|
\end{aligned}
$$

From the regularity of $f$, $g$, and their gradient given in Assumption 3.1, we obtain:

$$\|A(y_\Phi) - A(y^\star)\| \leq L_{g,2} \Delta_y(x, \xi)$$

$$\|B(y_\Phi)^{-1} - B(y^\star)^{-1}\| \leq \|B(y_\Phi)^{-1}\| \cdot \|B(y_\Phi) - B(y^\star)\| \cdot \|B(y^\star)^{-1}\| \leq \frac{L_{g,2}}{\mu^2} \Delta_y(x, \xi)$$

$$\|C(y_\Phi) - C(y^\star)\| \leq L_{f,1} \Delta_y(x, \xi)$$
$$\|A(y_\Phi)\| \leq L_{g,1},$$
$$\|A(y^\star)\| \leq L_{g,1},$$
$$\|C(y_\Phi)\| \leq L_{f,0},$$
$$\|C(y^\star)\| \leq L_{f,0},$$

where the second inequality uses the identity $\|X^{-1} - Y^{-1}\| \leq \|X^{-1}\| \cdot \|X - Y\| \cdot \|Y^{-1}\|$ and the fact that $\|B(y_\Phi)^{-1}\|$ and $\|B(y^\star)^{-1}\|$ are upper bounded by $1/\mu$ from the strong convexity of $g$.

It follows that:

$$
\begin{aligned}
\|\nabla F_{\Phi^\epsilon}(x, \xi, \eta) - \nabla F(x, \xi, \eta)\| &\leq L_{f,1} \Delta_y(x, \xi) + \frac{L_{g,2} L_{f,0}}{\mu} \Delta_y(x, \xi) \\
&\quad + \frac{L_{g,2} L_{g,1} L_{f,0}}{\mu^2} \Delta_y(x, \xi) + \frac{L_{f,1} L_{g,1} L_{g,2}}{\mu} \Delta_y(x, \xi) \\
&= K \Delta_y(x, \xi)
\end{aligned}
$$

where $K = L_{f,1} + \frac{L_{g,2} L_{f,0}}{\mu} + \frac{L_{g,2} L_{g,1} L_{f,0}}{\mu^2} + \frac{L_{f,1} L_{g,1}}{\mu}$.

We then bound the difference between the hypergradients using Jensen's inequality:

$$
\begin{aligned}
\|\nabla F_{\Phi^\epsilon}(x) - \nabla F(x)\| &\leq \mathbb{E}_{(\xi, \eta)} \|\nabla F_{\Phi^\epsilon}(x, \xi, \eta) - \nabla F(x, \xi, \eta)\| \\
&\leq K \cdot \mathbb{E}_{(\xi, \eta)} [\Delta_y(x, \xi)] \\
&= K \cdot \mathbb{E}_\xi \|y_\Phi(W^\star(x), \xi) - y^\star(x, \xi)\|
\end{aligned}
$$

$\square$

We then proceed with the proof of Proposition 4.2.

*Proof.* Let $x \in \mathbb{R}^{d_x}$, $W \in \mathbb{R}^{d_y \times N}$, and $G(x, y, \xi) \triangleq \mathbb{E}_{\eta \sim \mathbb{P}_{\eta|\xi}} [g(x, y, \xi, \eta)]$. As $g$ is $L_{g,1}$-smooth in $(x, y)$ for any $(\xi, \eta)$, and $\mu$-strongly convex in $y$ for any fixed $(x, \xi, \eta)$, so is $G$. Since $W^\star(x)$ minimizes $\mathbb{E}_\xi [G(x, y_\Phi(\cdot, \xi), \xi)]$, we have in particular:

$$\mathbb{E}_\xi [G(x, y_\Phi(W^\star(x), \xi), \xi)] \leq \mathbb{E}_\xi [G(x, y_\Phi(W, \xi), \xi)] \tag{6}$$

Additionally, since $y^\star(x, \xi)$ minimizes $G(x, \cdot, \xi)$, it also holds that:

$$G(x, y^\star(x, \xi), \xi) \leq G(x, y_\Phi(W^\star(x), \xi), \xi)$$

Using the strong convexity of $G$ at $W^\star(x)$ we thus obtain:

$$G(x, y_\Phi(W^\star(x), \xi), \xi) - G(x, y^\star(x, \xi), \xi) \geq \frac{\mu}{2} \|y_\Phi(W^\star(x), \xi) - y^\star(x, \xi)\|^2, \quad \forall \xi \in \Xi \quad (7)$$

On the other hand, the smoothness of $G$ yields:

$$|G(x, y_\Phi(W, \xi), \xi) - G(x, y^\star(x, \xi), \xi)| \leq L_{g,1} \|y_\Phi(W, \xi) - y^\star(x, \xi)\|^2, \quad \forall \xi \in \Xi \quad (8)$$

Taking the expectation over $\xi$ on both sides in equation 7 and equation 8, and using the inequality equation 6 we have:

$$\begin{aligned}
\mathbb{E}_\xi \|y_\Phi(W^\star(x), \xi) - y^\star(x, \xi)\|^2 &\leq \frac{2}{\mu} \mathbb{E}_\xi \left[ G(x, y_\Phi(W^\star(x), \xi), \xi) - G(x, y^\star(x, \xi), \xi) \right] \\
&\leq \frac{2}{\mu} \mathbb{E}_\xi \left[ G(x, y_\Phi(W, \xi), \xi) - G(x, y^\star(x, \xi), \xi) \right] \\
&\leq \frac{2L_{g,1}}{\mu} \mathbb{E}_\xi \|y_\Phi(W, \xi) - y^\star(x, \xi)\|^2
\end{aligned}$$

As this holds for any $x \in \mathbb{R}^{d_x}$, we obtain the desired result. $\qquad \square$

## C  REGULARITY OF EQUATION $\text{SBO}_{\Phi[N]}$

For readability, we refer in this section to $M_\Phi(\epsilon)$ as $M_\Phi$, and to $m_\Phi(\epsilon)$ as $m_\Phi$.

**Lemma C.1.** *If $A \succeq \mu I_{d_A}$ and $B \succeq 0$ with $\mu \geq 0$, then $A \otimes B \succeq \mu I_{d_A} \otimes B$ and $B \otimes A \succeq \mu B \otimes I_{d_A}$.*

*Proof.* By the bilinearity of the Kronecker product we have:

$$A \otimes B - \mu I_{d_A} \otimes B = (A - \mu I_{d_A}) \otimes B$$

Since the Kronecker product of two positive definite matrix is positive definite, it holds that $(A - \mu I_{d_A}) \otimes B \succeq 0$ and thus $A \otimes B - \mu A \otimes I_{d_B} \succeq 0$. With a symmetrical argument, we obtain $B \otimes A \succeq \mu B \otimes I_{d_A}$. $\qquad \square$

### C.1  STRONG CONVEXITY OF $G_{\Phi^\epsilon}(x, W) \triangleq \mathbb{E}_{(\xi, \eta)} \left[ g(x, W\Phi(\xi), \xi, \eta) \right]$

**Lemma C.2.** *Under assumptions 3.1-(ii) and if $\Phi$ is well-conditioned, $G_{\Phi^\epsilon}$ is $\mu m_\Phi(\epsilon)$-strongly convex in $W$ for any fixed $x \in \mathbb{R}^{d_x}$.*

*Proof.* We have for any fixed $x \in \mathbb{R}^{d_x}$ that $G_{\Phi^\epsilon}$ is twice differentiable with respect to $W$ as the expectation of compositions of twice differentiable and linear mapping. Additionally:

$$\nabla_{22}^2 G_{\Phi^\epsilon}(x, W) = \mathbb{E}_\xi \left[ \nabla_{22}^2 G(x, W\Phi(\xi), \xi) \otimes \Phi(\xi)\Phi(\xi)^\top \right]$$

Since $G$ is $\mu$-strongly convexity with respect to $y$ with $\mu > 0$ and $\mathbb{E}_\xi \left[ \Phi(\xi)\Phi(\xi)^\top \right] \succeq m_\Phi(\epsilon) I_{N_\Phi(\epsilon)} \succeq 0$, we obtain using Lemma C.1 twice:

$$\begin{aligned}
\nabla_{22}^2 G_{\Phi^\epsilon}(x, W) &\succeq \mu \mathbb{E}_\xi \left[ I_{d_y} \otimes \Phi(\xi)\Phi(\xi)^\top \right] \\
&= \mu I_{d_y} \otimes \mathbb{E}_\xi \left[ \Phi(\xi)\Phi(\xi)^\top \right] \\
&\succeq \mu m_\Phi(\epsilon) I_{d_y} \otimes I_{N_\Phi(\epsilon)} \\
&= \mu m_\Phi(\epsilon) I_{d_y \cdot N_\Phi(\epsilon)}
\end{aligned}$$

and we conclude that $G_{\Phi^\epsilon}(x, W)$ is $\mu m_\Phi(\epsilon)$-strongly convex with respect to $W$ for any $x \in \mathbb{R}_{d_x}$. $\quad \square$

### C.2  LIPSCHITZ CONTINUITY

**Lemma C.3.** *Let $h(x, y, \xi, \eta) : \mathbb{R}^{d_x} \times \mathbb{R}^{d_y} \times \Xi \times \mathbb{R}^{d_\eta} \to \mathbb{R}^n$ and $l(\xi) : \Xi \to \mathbb{R}^N$. Suppose the following conditions hold:*

    *1. $h$ is $L$-Lipschitz continuous with respect to $(x, y)$ for any fixed $(\xi, \eta)$.*

> 2. *There exists a constant $M$ such that:*

$$\|l(\xi)\| \leq M, \quad \forall \xi \in \Xi$$

*Then the mapping $h_\Phi : (x, W, \xi, \eta) \mapsto h(x, W\Phi(\xi), \xi, \eta)l(\xi)$ is $LM_\Phi M$-Lipschitz continuous with respect to $(x, y)$ for all fixed $(\xi, \eta)$.*

*Proof.* Since $h(x, y, \xi, \eta)$ is $L$-Lipschitz continuous in $(x, y)$ and $\sup_{\xi \in \Xi} \|\Phi(\xi)\| \leq M_\Phi$, we have for any $(x, W)$ and $(x', W')$:

$$\|h_\Phi(x, W, \xi, \eta) - h_\Phi(x', W', \xi, \eta)\| = \|h(x, W\Phi(\xi), \xi, \eta)l(\xi) - h(x', W'\Phi(\xi), \xi, \eta)l(\xi)\|$$
$$\overset{(1)}{\leq} \|h(x, W\Phi(\xi), \xi, \eta) - h(x', W'\Phi(\xi), \xi, \eta)\| \cdot \|l(\xi)\|$$
$$\overset{(2)}{\leq} ML\left(\|x - x'\| + \|W\Phi(\xi) - W'\Phi(\xi)\|\right)$$
$$\overset{(3)}{\leq} ML\left(\|x - x'\| + M_\Phi\|W - W'\|\right)$$
$$\overset{(4)}{\leq} LM_\Phi M\left(\|x - x'\| + \|W - W'\|\right)$$

where (1) uses the sub-multiplicativity of $\|\cdot\|$, (2) follows from the Lipschitz continuity of $h$ and this inequality $\|l(\xi)\| \leq M$, (3) uses again the sub-multiplicativity of $\|\cdot\|$ and the inequality $\|\Phi(\xi)\| \leq M_\Phi(\epsilon)$, and (4) holds since $1 \leq M_\Phi$. Therefore $h_\Phi$ is $LM_\Phi M$-Lipschitz continuous in $(x, W)$. $\square$

**Lemma C.4.** *The following hold under assumption 3.1:*

> 1. *The functions $f_{\Phi^\epsilon}(x, W, \xi, \eta)$ is $L_{f,0}M_\Phi$-Lipschitz with respect to $x$ and $W$.*
>
> 2. *The gradients $\nabla f_{\Phi^\epsilon}(x, W, \xi, \eta)$ and $\nabla g_{\Phi^\epsilon}(x, W, \xi, \eta)$ are $L_{f,1}M_\Phi^2$ and $L_{g,1}M_\Phi^2$-Lipschitz, respectively, with respect to $x$ and $W$.*
>
> 3. *The second order gradients $\nabla_{12}^2 g_{\Phi^\epsilon}(x, W, \xi, \eta)$ and $\nabla_{22}^2 g_{\Phi^\epsilon}(x, W, \xi, \eta)$ are $L_{g,2}M_\Phi^3$-Lipschitz with respect to $x$ and $W$.*

*Proof.* Under assumption 3.1-(i), the mappings $f$, $\nabla f$, $\nabla g$, and $\nabla^2 g$, are $L_{f,0}$, $L_{f,1}$, $L_{g,1}$, and $L_{g,2}$-Lipschitz continuous with respect to $(x, y)$ for any fixed $(\xi, \eta)$, respectively. Additionally, the chain rule gives:

$$\nabla_1 f_{\Phi^\epsilon}(x, W, \xi, \eta) = \nabla_1 f(x, W\Phi(\xi), \xi, \eta)$$
$$\nabla_2 f_{\Phi^\epsilon}(x, W, \xi, \eta) = \nabla_2 f(x, W\Phi(\xi), \xi, \eta)\Phi(\xi)^\top$$
$$\nabla_1 g_{\Phi^\epsilon}(x, W, \xi, \eta) = \nabla_1 g(x, W\Phi(\xi), \xi, \eta)$$
$$\nabla_2 g_{\Phi^\epsilon}(x, W, \xi, \eta) = \nabla_2 g(x, W\Phi(\xi), \xi, \eta)\Phi(\xi)^\top$$
$$\nabla_{12}^2 g_{\Phi^\epsilon}(x, W, \xi, \eta) = \nabla_{12}^2 g(x, W\Phi(\xi), \xi, \eta)\Phi(\xi)^\top$$
$$\nabla_{22}^2 g_{\Phi^\epsilon}(x, W, \xi, \eta) = \nabla_{22}^2 g(x, W\Phi(\xi), \xi, \eta) \otimes \Phi(\xi)\Phi(\xi)^\top$$

We can then use Lemma C.3 to obtain the following:

| $h$ | $l$ | $L$ | $M$ | Lipschitz coefficient of $h(x, W\Phi(\xi), \xi, \eta)$ w.r.t. $(x, W)$ |
|:---:|:---:|:---:|:---:|:---:|
| $f_{\Phi^\epsilon}$ | $1$ | $L_{f,0}$ | $1$ | $L_{f,0}M_\Phi$ |
| $\nabla_1 f_{\Phi^\epsilon}$ | $I_{d_x}$ | $L_{f,1}$ | $1$ | $L_{f,1}M_\Phi$ |
| $\nabla_2 f_{\Phi^\epsilon}$ | $\Phi^\top$ | $L_{f,1}$ | $M_\Phi$ | $L_{f,1}M_\Phi^2$ |
| $\nabla_1 g_{\Phi^\epsilon}$ | $I_{d_x}$ | $L_{g,1}$ | $1$ | $L_{g,1}M_\Phi$ |
| $\nabla_2 g_{\Phi^\epsilon}$ | $\Phi^\top$ | $L_{g,1}$ | $M_\Phi$ | $L_{g,1}M_\Phi^2$ |
| $\nabla_{12}^2 g_{\Phi^\epsilon}$ | $\Phi^\top$ | $L_{g,2}$ | $M_\Phi$ | $L_{g,2}M_\Phi^2$ |
| $\nabla_{22}^2 g_{\Phi^\epsilon}$ | $\Phi\Phi^\top$ | $L_{g,2}$ | $M_\Phi^2$ | $L_{g,2}M_\Phi^3$ |

and we conclude using the inequality $1 \leq M_\Phi$. $\qquad\square$

## C.3 BOUNDED VARIANCE

**Lemma C.5.** *Under assumption 3.1, then for all $\xi$, $y^\star(\cdot, \xi)$ is $L_y$-Lipschitz continuous with:*

$$L_y = \frac{L_{g,1}}{\mu}$$

*Proof.* Let $g(x, y, \xi) = \mathbb{E}_{\eta|\xi} g(x, W, \xi, \eta)$. By definition of $y^\star$, we have $\nabla_2 g(x, y^\star(x, \xi), \xi) = 0$. Then, by taking the derivative on both sides w.r.t. $x$, using the chain rule, and the implicit function theorem, we obtain:

$$\nabla_{12}^2 g(x, y^\star(x), \xi) + \nabla_{22}^2 g(x, y^\star(x, \xi), \xi) \nabla_1 y^\star(x, \xi) = 0$$

It follows that $\|\nabla_1 y^\star(x, \xi)\| \leq \frac{L_{g,1}}{\mu}$. $\qquad\square$

**Lemma C.6.** *Let $h(x, y, \xi, \eta) : \mathbb{R}^{d_x} \times \mathbb{R}^{d_y} \times \Xi \times \mathbb{R}^{d_\eta} \to \mathbb{R}^n$ and $l(\xi) : \Xi \to \mathbb{R}^N$. Suppose the following conditions hold:*

1. *$h(x, y, \xi, \eta)$ has, conditioned on $\xi$, a variance bounded by $\sigma^2$ i.e.*

$$\mathbb{E}_{\eta \sim \mathbb{P}_{\eta|\xi}} \left\| h(x, y, \xi, \eta) - \mathbb{E}_{\eta' \sim \mathbb{P}_{\eta|\xi}} [h(x, y, \xi, \eta')] \right\|^2 \leq \sigma^2, \quad \forall x \in \mathbb{R}^{d_x}, y \in \mathbb{R}^{d_y}, \xi \in \mathbb{R}^{d_\xi}.$$

2. *There exists a constant $C$ such that:*

$$\|h(x, y, \xi, \eta)\| \leq C, \quad \forall x \in \mathbb{R}^{d_x}, y \in \mathbb{R}^{d_y}, \xi \in \mathbb{R}^{d_\xi}$$

3. *There exists a constant $M$ such that:*

$$\|l(\xi)\| \leq M, \quad \forall \xi \in \Xi$$

*Then the mapping $h_\Phi : (x, W, \xi, \eta) \mapsto h(x, W\Phi(\xi), \xi, \eta) l(\xi)$ has a variance bounded by $M^2 (\sigma^2 + C^2)$.*

*Proof.* Let $Z = h_\Phi(x, W, \xi, \eta)$. Using the law of total variance we have:

$$\text{Var}_{(\xi,\eta)} [Z] = \mathbb{E}_\xi \left[ \text{Var}_{\eta|\xi} [Z \mid \xi] \right] + \text{Var}_\xi \left[ \mathbb{E}_{\eta|\xi} [Z \mid \xi] \right]$$

For all $\xi \in \Xi$ it holds that:

$$\text{Var}_{\eta|\xi} [Z \mid \xi] = \mathbb{E}_{\eta|\xi} \left[ \left\| h(x, W\Phi(\xi), \xi, \eta) l(\xi) - \mathbb{E}_{\eta'|\xi} [h(x, W\Phi(\xi), \xi, \eta') l(\xi)] \right\|^2 \Big| \xi \right]$$

$$\leq \|l(\xi)\|^2 \cdot \mathbb{E}_{\eta|\xi} \left[ \left\| h(x, W\Phi(\xi), \xi, \eta) - \mathbb{E}_{\eta'|\xi} [h(x, W\Phi(\xi), \xi, \eta')] \right\|^2 \Big| \xi \right]$$

$$\leq \|l(\xi)\|^2 \cdot \sigma^2$$

where the second inequality follows from the sub-multiplicativity of $\| \cdot \|$ and the fact that $l(\xi)$ is deterministic when conditioned on $\xi$, and the last inequality holds under condition 1.

Similarly, we have that for all $\xi \in \Xi$ and under condition 2:

$$\left\| \mathbb{E}_{\eta|\xi} [Z \mid \xi] \right\| \leq \left\| \mathbb{E}_{\eta|\xi} [h(x, W\Phi(\xi), \xi, \eta) l(\xi) \mid \xi] \right\|$$

$$\leq \|l(\xi)\| \cdot \left\| \mathbb{E}_{\eta|\xi} [h(x, W, \xi, \eta) \mid \xi] \right\|$$

$$\leq \|l(\xi)\| \cdot C$$

Combining the above results, we obtain:

$$\text{Var}_{(\xi,\eta)} [Z] \leq \mathbb{E}_\xi \left[ \|l(\xi)\|^2 \sigma^2 \right] + \mathbb{E}_\xi \left[ \left\| \mathbb{E}_{\eta|\xi} [Z \mid \xi] \right\|^2 \right]$$

$$\leq M^2 \sigma^2 + \mathbb{E}_\xi \left[ \|l(\xi)\|^2 \cdot C^2 \right]$$

$$\leq M^2 (\sigma^2 + C^2)$$

where the last inequality uses condition 3. $\qquad\square$

**Lemma C.7.** *Under assumption 3.1, the mappings* $\nabla_1 f_{\Phi^\epsilon}(x, W, \xi, \eta)$, $\nabla_2 f_{\Phi^\epsilon}(x, W, \xi, \eta)$, $\nabla_2 g_{\Phi^\epsilon}(x, W, \xi, \eta)$, $\nabla_{12}^2 g_{\Phi^\epsilon}(x, W, \xi, \eta)$, *and* $\nabla_{22}^2 g_{\Phi^\epsilon}(x, W, \xi, \eta)$ *have a bounded variance for all* $x$ *and* $W$ *such that* $\|W - W^\star(x)\| \leq \Delta_0$.

*Proof.* Under assumption 3.1-(iii) we have for any $(x, y, \xi)$ that the variances of $\nabla_1 f(x, y, \xi, \eta)$ and $\nabla_2 f(x, y, \xi, \eta)$ are upper bounded by $\sigma_f^2$, and the variance of $\nabla_{12}^2 g(x, y, \xi, \eta)$ and $\nabla_{22}^2 g(x, y, \xi, \eta)$ by $\sigma_{g,2}^2$. Thus these 4 functions satisfy the first condition of Lemma C.7. From the Lipschitz continuity of $f$ and $\nabla g$ (assumption 3.1-(i)) we have for any $(x, y, \xi, \eta)$:

$$\|\nabla_1 f(x, y, \xi, \eta)\| \leq L_{f,0}$$
$$\|\nabla_2 f(x, y, \xi, \eta)\| \leq L_{f,0}$$
$$\left\|\nabla_{12}^2 g(x, y, \xi, \eta)\right\| \leq L_{g,1}$$
$$\left\|\nabla_{22}^2 g(x, y, \xi, \eta)\right\| \leq L_{g,1}$$

Taking the expectation over $\eta \sim \mathbb{P}_{\eta|\xi}$, the second condition of Lemma C.7 holds for theses 4 functions. We then use the chain rule to get:

$$\nabla_1 f_{\Phi^\epsilon}(x, W, \xi, \eta) = \nabla_1 f(x, W\Phi(\xi), \xi, \eta)$$
$$\nabla_2 f_{\Phi^\epsilon}(x, W, \xi, \eta) = \nabla_2 f(x, W\Phi(\xi), \xi, \eta)\Phi(\xi)^\top$$
$$\nabla_{12}^2 g_{\Phi^\epsilon}(x, W, \xi, \eta) = \nabla_{12}^2 g(x, W\Phi(\xi), \xi, \eta)\Phi(\xi)^\top$$
$$\nabla_{22}^2 g_{\Phi^\epsilon}(x, W, \xi, \eta) = \nabla_{22}^2 g(x, W\Phi(\xi), \xi, \eta) \otimes \Phi(\xi)\Phi(\xi)^\top$$

Since $\|\Phi(\xi)\| \leq M_\Phi$, Lemma C.6 applies and we obtain the following bounds:

| $h$ | $\sigma^2$ | $l$ | $C$ | $M$ | Bound on $\mathrm{Var}_{(\xi,\eta)}\left[h(x, W, \xi, \eta)\right]$ |
|---|---|---|---|---|---|
| $\nabla_1 f_{\Phi^\epsilon}$ | $\sigma_f^2$ | $I_{d_x}$ | $L_{f,0}$ | $1$ | $\sigma_f^2 + L_{f,0}^2$ |
| $\nabla_2 f_{\Phi^\epsilon}$ | $\sigma_f^2$ | $\Phi^\top$ | $L_{f,0}$ | $M_\Phi$ | $M_\Phi^2\left(\sigma_f^2 + L_{f,0}^2\right)$ |
| $\nabla_{12}^2 g_{\Phi^\epsilon}$ | $\sigma_{g,2}^2$ | $\Phi^\top$ | $L_{g,1}$ | $M_\Phi$ | $M_\Phi^2\left(\sigma_{g,2}^2 + L_{g,1}^2\right)$ |
| $\nabla_{22}^2 g_{\Phi^\epsilon}$ | $\sigma_{g,2}^2$ | $\Phi\Phi^\top$ | $L_{g,1}$ | $M_\Phi^2$ | $M_\Phi^4\left(\sigma_{g,2}^2 + L_{g,1}^2\right)$ |

We conclude that these 4 mappings have a bounded variance.

The case of $\nabla_2 g_{\Phi^\epsilon}$ requires extra care since it is not uniformly bounded. We have:

$$\|\nabla_2 g(x, W\Phi(\xi), \xi, \eta)\| \leq \|\nabla_2 g(x, W\Phi(\xi), \xi, \eta) - \nabla_2 g(x, y^\star(x, \xi), \xi, \eta)\| + \|\nabla_2 g(x, y^\star(x, \xi), \xi, \eta)\|$$
$$\leq L_{g,1}\|W\Phi(\xi) - y^\star(x, \xi)\| + \|\nabla_2 g(x, y^\star(x, \xi), \xi, \eta)\|$$

Using the identity $(a + b)^2 \leq 2a^2 + 2b^2$, we have:

$$\|\nabla_2 g(x, W\Phi(\xi), \xi, \eta)\|^2 \leq 2L_{g,1}^2\|W\Phi(\xi) - y^\star(x, \xi)\|^2 + 2\|\nabla_2 g(x, y^\star(x, \xi), \xi, \eta)\|^2$$

We use the triangular inequality, the identity $(a + b)^2 \leq 2a^2 + 2b^2$, and $\|\Phi(\xi)\| \leq M_\Phi$ to bound:

$$\|W\Phi(\xi) - y^\star(x, \xi)\|^2 \leq \left(\|W\Phi(\xi) - W^\star(x)\Phi(\xi)\| + \|W^\star(x)\Phi(\xi) - y^\star(x, \xi)\|\right)^2$$
$$\leq 2\|W\Phi(\xi) - W^\star(x)\Phi(\xi)\|^2 + 2\|W^\star(x)\Phi(\xi) - y^\star(x, \xi)\|^2$$
$$\leq 2M_\Phi^2\|W - W^\star(x)\|^2 + 2\|W^\star(x)\Phi(\xi) - y^\star(x, \xi)\|^2$$

Finally, combining $\|W - W^\star(x)\| \leq \Delta_0$, equation 3, and equation 1 yields:

$$\mathbb{E}_\xi\|W\Phi(\xi) - y^\star(x, \xi)\|^2 \leq 2M_\Phi^2\Delta_0^2 + 2\|W^\star(x)\Phi(\xi) - y^\star(x, \xi)\|^2$$
$$\leq 2M_\Phi^2\Delta_0^2 + \frac{4L_{g,1}}{\mu}\mathbb{E}_\xi\|W^\dagger(x)\Phi(\xi) - y^\star(x, \xi)\|^2$$
$$\leq 2M_\Phi^2\Delta_0^2 + \frac{\epsilon^2}{K^2}$$

From Assumption 3.1, for any fixed $\xi$, $\nabla_2 g(x, y, \xi, \eta)$ is unbiased with variance bounded by $\sigma_{g,1}^2$. By definition of $y^\star(x, \xi)$ we have $\mathbb{E}_{\eta|\xi} \nabla_2 g(x, y^\star(x, \xi), \xi, \eta) = 0$ for all $\xi$ and it follows that:

$$\mathbb{E}_{\eta|\xi} \|\nabla_2 g(x, y^\star(x, \xi), \xi, \eta)\|^2 = \text{Var}_{\eta|\xi} \left[\nabla_2 g(x, y^\star(x, \xi), \xi, \eta)\right]$$
$$\leq \sigma_{g,1}^2$$

Taking the expectation over $\xi$ yields:

$$\mathbb{E}_{(\xi, \eta)} \|\nabla_2 g(x, y^\star(x, \xi), \xi, \eta)\|^2 \leq \sigma_{g,1}^2$$

Combining the above results we have:

$$\mathbb{E}_{(\xi, \eta)} \|\nabla_2 g(x, W\Phi(\xi), \xi, \eta)\|^2 \leq 2L_{g,1}^2 \mathbb{E}_{(\xi, \eta)} \|W\Phi(\xi) - y^\star(x, \xi)\|^2 + 2\mathbb{E}_{(\xi, \eta)} \|\nabla_2 g(x, y^\star(x, \xi), \xi, \eta)\|^2$$
$$\leq 2L_{g,1}^2 \left(2M_\Phi^2 \Delta_0^2 + \frac{\epsilon^2}{K^2}\right) + 2\sigma_{g,1}^2$$

Since $\nabla_2 g_{\Phi^\epsilon}(x, W, \xi, \eta) = \nabla_2 g(x, W\Phi(\xi), \xi, \eta)\Phi(\xi)^\top$ and $\|\Phi(\xi)\| \leq M_\Phi$ we obtain:

$$\text{Var}_{(\xi, \eta)} \left[\nabla_2 g_{\Phi^\epsilon}(x, W, \xi, \eta)\right] \leq \mathbb{E}_{(\xi, \eta)} \|\nabla_2 g_{\Phi^\epsilon}(x, W, \xi, \eta)\|^2 \quad\quad (9)$$
$$\leq \mathbb{E}_{(\xi, \eta)} \left[\|\nabla_2 g(x, W\Phi(\xi), \xi, \eta)\|^2 \|\Phi(\xi)\|^2\right]$$
$$\leq 4L_{g,1}^2 M_\Phi^4 \Delta_0^2 + 2L_{g,1}^2 M_\Phi^2 \frac{\epsilon^2}{K^2} + 2\sigma_{g,1}^2 M_\Phi^2$$
$$\leq 4M_\Phi^4 \left(L_{g,1}^2 \Delta_0^2 + \frac{L_{g,1}^2 \epsilon^2}{K^2} + \sigma_{g,1}^2\right)$$

which concludes the proof. $\quad\quad\square$

## D    PROOF OF THEOREM 4.4

In this proof, we use the notations $\lesssim$, $\simeq$, and $\gtrsim$ to denote relations up to a constant. We will show that under assumptions 3.1 and if $\Phi$ is well-conditioned, the assumptions 1 and 2 of Guo et al. (2021) hold for $f_\Phi$ and $g_\Phi$. Namely, we want to show:

**Assumption D.1.** For any $x \in \mathbb{R}^{d_x}$, $\mathbb{E}_{(\xi, \eta)} \left[g_\Phi(x, W, \xi, \eta)\right]$ is $\lambda$-strongly convex and $L$-smooth.

and

**Assumption D.2.** The following hold:

(i) $\nabla_1 f_\Phi$ is $L_{fx}$-Lipschitz continuous, $\nabla_2 f_\Phi$ is $L_{fy}$-Lipschitz continuous, $\nabla_2 g_\Phi$ is $L_{gy}$-Lipschitz continuous, $\nabla_{12}^2 g_\Phi$ is $L_{gxy}$-Lipschitz continuous, $\nabla_{22}^2 g_\Phi$ is $L_{gyy}$-Lipschitz continuous, all with respect to $(x, W)$.

(ii) $\nabla_1 f_\Phi$, $\nabla_2 f_\Phi$, $\nabla_2 g_\Phi$, $\nabla_{12}^2 f_\Phi$, and $\nabla_{22}^2 f_\Phi$ have a variance bounded by $\sigma^2$.

(iii) $\|\nabla_2 \mathbb{E}_{(\xi, \eta)} \left[f(x, W, \xi, \eta)\right]\|^2 \leq C_{fy}^2$, $\|\nabla_{12}^2 \mathbb{E}_{(\xi, \eta)} \left[g(x, W, \xi, \eta)\right]\|^2 \leq C_{gxy}^2$.

Under assumption 3.1 and if $\Phi$ is well-conditioned, Lemma C.2 gives $\mathbb{E}_{(\xi, \eta)} \left[g_\Phi(x, W, \xi, \eta)\right]$ is $\mu m_\Phi(\epsilon)$-strongly convex with respect to $W$ for any $x \in \mathbb{R}^{d_x}$. Further, we have from Lemma C.4 that $\nabla_2 g_\Phi(x, W, \xi, \eta)$ is $L_{g,1} M_\Phi^2$-Lipchitz continuous in $W$ for all fixed $(x, \xi, \eta)$. Taking the expectation over $(\xi, \eta)$ we obtain:

$$\left\|\nabla_2 \mathbb{E}_{(\xi, \eta)} \left[g_\Phi(x, W, \xi, \eta)\right] - \nabla_2 \mathbb{E}_{(\xi, \eta)} \left[g_\Phi(x', W', \xi, \eta)\right]\right\|$$
$$= \left\|\mathbb{E}_{(\xi, \eta)} \left[\nabla_2 g_\Phi(x, W, \xi, \eta) - \nabla_2 g_\Phi(x', W', \xi, \eta)\right]\right\|$$
$$\leq \mathbb{E}_{(\xi, \eta)} \left[\|\nabla_2 g_\Phi(x, W, \xi, \eta) - \nabla_2 g_\Phi(x', W', \xi, \eta)\|\right]$$
$$\leq L_{g,1} M_\Phi^2 \left(\|x - x'\| + \|W - W'\|\right)$$

where the first inequality holds since assumption 3.1 implies that $\nabla_2 g_\Phi(x, W, \xi, \eta)$ is unbiased and the first inequality uses Jensen's inequality. Therefore $\mathbb{E}_{(\xi, \eta)} \left[g_\Phi(x, W, \xi, \eta)\right]$ is $L_{g,1} M_\Phi^2$-smooth for any fixed $x \in \mathbb{R}^{d_x}$ and assumption D.1 holds with $\lambda = \mu m_\Phi(\epsilon)$ and $L = L_{g,1} M_\Phi^2$.

Using the implicit-function theorem, the accuracy of the estimate of $\nabla F_\Phi$ degrades linearly with the gap $\|W_t - W^\star(x_t)\|$. Hence $\|W_t - W^\star(x_t)\|$ cannot diverge and there exists $\Delta_0 < \infty$ such that the iterates $W_t$ satisfy $\|W_t - W^\star(x_t)\| \leq \Delta_0$. In particular, this is holds for RSVRB as per Lemma 6 of Guo et al. (2021). Henceforth, we restrict our analysis to $\left\{ (x, W) \in \mathbb{R}^{d_x} \times \mathbb{R}^{d_y \times N_\Phi(\epsilon)} : \|W - W^\star(x)\| \leq \Delta_0 \right\}$ for the rest of the proof.

Lemma C.4 and Lemma C.7 show that assumption $D.2 - (i)$ and $D.2 - (ii)$ hold $f_\Phi$ and $g_\Phi$ for with Lipchitz constant $L$ and uniform variance bound $\sigma^2$ given in the table below.

| $h$ | $L$ | $\sigma^2$ |
|---|---|---|
| $\nabla_1 f_{\Phi^\epsilon}$ | $L_{f,1} M_\Phi$ | $\sigma_f^2 + L_{f,0}^2$ |
| $\nabla_2 f_{\Phi^\epsilon}$ | $L_{f,1} M_\Phi^2$ | $M_\Phi^2 \left( \sigma_f^2 + L_{f,0}^2 \right)$ |
| $\nabla_2 g_{\Phi^\epsilon}$ | $L_{g,1} M_\Phi^2$ | $4 M_\Phi^4 \left( L_{g,1}^2 \Delta_0^2 + \frac{\epsilon}{K^2} + \sigma_{g,1}^2 \right)$ |
| $\nabla_{12}^2 g_{\Phi^\epsilon}$ | $L_{g,2} M_\Phi^2$ | $M_\Phi^2 \left( \sigma_{g,2}^2 + L_{g,1}^2 \right)$ |
| $\nabla_{22}^2 g_{\Phi^\epsilon}$ | $L_{g,2} M_\Phi^3$ | $M_\Phi^4 \left( \sigma_{g,2}^2 + L_{g,1}^2 \right)$ |

Further, Lemma C.4 also gives that $f_\Phi$ and $g_\phi$ are $L_{f,0} M_\Phi$ and $L_{g,0} M_\Phi$-Lipschitz continuous in $(x, W)$, respectively. It follows that that $\|\nabla_1 \mathbb{E}_{(\xi,\eta)} [f_\Phi (x, W, \xi, \eta)]\| \leq L_{f,0} M_\Phi$ and $\|\nabla_2 \mathbb{E}_{(\xi,\eta)} [g_\Phi (x, W, \xi, \eta)]\| \leq L_{g,0} M_\Phi$. Hence condition $D.2 - (iii)$ hold for $f_\Phi$ and $g_\Phi$ with $C_{fy} = L_{f,0} M_\Phi$ and $C_{gxy} = L_{g,1} M_\Phi^2$.

Therefore Theorem 1 of Guo et al. (2021) holds. Before stating it we first bound some quantities in term of $M_\Phi$ and $m_\Phi$.

Since $\Phi$ is well conditioned, we have $\Sigma_\Phi \succeq m_\phi I_{N_\Phi(\epsilon)}$. Therefore:

$$
\begin{aligned}
\mathbb{E}_\xi \left[ \|W^\star(x_0)\Phi(\xi)\|^2 \right] &= \mathbb{E}_\xi \left[ (W^\star(x_0)\Phi(\xi))^\top W^\star(x_0)\Phi(\xi) \right] \\
&= \mathbb{E}_\xi \left[ \text{tr} \left( W^\star(x_0)\Phi(\xi)(W^\star(x_0)\Phi(\xi))^\top \right) \right] \\
&= \text{tr} \left( W^\star(x_0)\mathbb{E}_\xi \left[ \Phi(\xi)\Phi(\xi)^\top \right] W^\star(x_0)^\top \right) \\
&\geq m_\phi \text{tr} \left( W^\star(x_0) W^\star(x_0)^\top \right) \\
&= m_\phi \|W^\star(x_0)\|_F^2 \\
&\geq m_\phi \|W^\star(x_0)\|^2
\end{aligned}
$$

For any $(x_0, W_0)$ and by the $\mu$ strong convexity of $g$ we have:

$$
\mathbb{E}_{(\xi,\eta)} [g_\Phi(x_0, W_0, \xi, \eta)] \geq \mathbb{E}_{(\xi,\eta)} [g_\Phi(x_0, W^\star(x_0), \xi, \eta)] + \frac{\mu}{2} \mathbb{E}_{(\xi,\eta)} \left[ \|W^\star(x_0)\Phi(\xi)\|^2 \right]
$$

and thus, taking $W_0 = 0$:

$$
\begin{aligned}
\mathbb{E}_\xi \left[ \|W^\star(x_0)\Phi(\xi)\|^2 \right] &\leq \frac{2}{\mu} \mathbb{E}_{(\xi,\eta)} [g_\Phi(x_0, W_0, \xi, \eta) - g_\Phi(x_0, W^\star(x_0), \xi, \eta)] \\
&\leq \frac{2}{\mu} \underbrace{\left( \mathbb{E}_{(\xi,\eta)} \left[ g(x_0, 0, \xi, \eta) - \min_y g(x_0, y, \xi, \eta) \right] \right)}_{\Delta_{g,0}}
\end{aligned}
$$

Combining the above results we obtain:

$$
\begin{aligned}
\|W^\star(x_0)\|^2 &\leq \frac{\mathbb{E}_\xi \left[ \|W^\star(x_0)\Phi(\xi)\|^2 \right]}{m_\Phi} \\
&\leq \frac{2\Delta_{g,0}}{\mu \cdot m_\Phi}
\end{aligned}
$$

From Lemma 2.2 in Ghadimi & Wang (2018), $W^\star(x)$ is Lipschitz continuous in $x$ with constant $L_W = \frac{L_{gy}}{\lambda} = \frac{L_{g,1} M_\Phi^2}{\mu m_\Phi}$. Additionally, $\nabla F$ is Lipschitz continuous in $x$ with constant

$$
\begin{aligned}
L_F &= L_{fy} + \frac{L_{fy} L_{gy}}{\lambda} + \frac{L_{gy}}{\lambda}\left(L_{fy} + \frac{L_{fy} L_{gy}}{\lambda} + L_f\left[\frac{L_{gxy}}{\lambda} + \frac{L_{gyy} L_{gy}}{\lambda^2}\right]\right) + L_f\left[\frac{L_{gxy} L_f}{\lambda} + \frac{L_{gyy} L_{gy}}{\lambda^2}\right] \\
&= L_{fy} + \frac{2 L_{fy} L_{gy} + L_f^2 L_{gxy}}{\lambda} + \frac{L_{fy} L_{gy}^2 + L_f L_{gy} L_{gxy} + L_f L_{gyy} L_{gy}}{\lambda^2} + \frac{L_f L_{gy}^2 L_{gyy}}{\lambda^3} \\
&\lesssim M_\Phi^2 + \frac{M_\Phi^5}{m_\Phi} + \frac{M_\Phi^6}{m_\Phi^2} + \frac{M_\Phi^8}{m_\Phi^3} \\
&\lesssim \frac{M_\Phi^8}{m_\Phi^3}
\end{aligned}
$$

Using the notations of Guo et al. (2021), we have $\delta_{W,0} \triangleq \|W_1 - W^\star(x_0)\|^2$ with $W_1 = W_0 - \tau_0 \tau w_1$ and:

$$
\begin{aligned}
\mathbb{E}\left[\delta_{fx,0}\right] &\lesssim \|\nabla_1 f_{\Phi^\epsilon}(x_0, W_0))\|^2 \lesssim M_\Phi^2 \\
\mathbb{E}\left[\delta_{fy,0}\right] &\lesssim \|\nabla_2 f_{\Phi^\epsilon}(x_0, W_0))\|^2 \lesssim M_\Phi^4 \\
\mathbb{E}\left[\delta_{gxy,0}\right] &\lesssim \|\nabla_{12} g_{\Phi^\epsilon}(x_0, W_0))\|^2 \lesssim M_\Phi^4 \\
\mathbb{E}\left[\delta_{gyy,0}\right] &\lesssim \|\nabla_{22} g_{\Phi^\epsilon}(x_0, W_0))\|^2 \lesssim M_\Phi^6 \\
\mathbb{E}\left[\delta_{gy,0}\right] &\lesssim \mathbb{E}\|\nabla_2 g_{\Phi^\epsilon}(x_0, W_0))\|^2 \lesssim M_\Phi^4
\end{aligned}
$$

where the first 4 inequalities follows from the Lipschitz constants given in Lemma C.4, and the last holds given equation 9 .

Theorem 1 of Guo et al. (2021) finally give:

$$
\begin{aligned}
\frac{1}{2(T+1)}\mathbb{E}\left[\sum_{t=0}^{T}\|\nabla F_{\Phi^\epsilon}(x_t)\|^2\right] \le{}& \frac{F_\Phi(x_0) - F_\Phi(x^\star)}{\gamma \eta_T T} + \frac{C\mathbb{E}\left[\delta_{W,0}\right]}{\eta_T T} \\
&+ \frac{\mathbb{E}\left[\delta_{gy,0} + \delta_{fx,0} + \delta_{fy,0} + \delta_{gxy,0} + \delta_{gyy,0}\right]}{\gamma \eta_0 \eta_T T} + \frac{\mathcal{O}\left(\ln(T+2)\right)}{\gamma \eta_T T}
\end{aligned}
\tag{10}
$$

where for $c = 1$:

$$C_0 = \left(2L_{fx}^2 + \frac{6C_{fy}^2 L_{gxy}^2}{\lambda^2} + \frac{6C_{fy}^2 C_{gxy}^2 L_{gyy}^2}{\lambda^4} + \frac{6L_{fy}^2 C_{gxy}^2}{\lambda^2}\right) \lesssim \frac{M_\Phi^{12}}{m_\Phi^4}$$

$$C_1 = 2$$

$$C_2 = \frac{6C_{fy}^2}{\lambda^2} \simeq \frac{M_\Phi^2}{m_\Phi^2}$$

$$C_3 = \frac{6C_{fy}^2 C_{gxy}^2}{\lambda^4} \simeq \frac{M_\Phi^6}{m_\Phi^4}$$

$$C_4 = \frac{6C_{gxy}^2}{\lambda^2} \simeq \frac{M_\Phi^4}{m_\Phi^2}$$

$$\tau = \frac{1}{3L_g} \simeq \frac{1}{M_\Phi^2}$$

$$C = \max\left\{\frac{4C_0}{\tau\lambda}, \frac{4\left(L_{gy}^2 + L_{fx}^2 + L_{fy}^2 + L_{gxy}^2 + L_{gyy}^2\right)}{\gamma}\right\} \lesssim \frac{M_\Phi^{18}}{m_\Phi^5}$$

$$\gamma = \min\left\{\frac{\sqrt{\tau\lambda}}{8\sqrt{C}L_W}, \frac{1}{16\left(L_{gy}^2 + L_{fx}^2 + L_{fy}^2 + L_{gxy}^2 + L_{gyy}^2\right)}\right\} \gtrsim \frac{m_\Phi^4}{M_\Phi^{12}}$$

$$c_0 = \max\left\{2, 64L_F^3, \left(\frac{\lambda}{16C\gamma\tau}\right)^{3/2}, \left(\frac{2}{7L_F}\right)^{3/2}, (2(C_1 + C_2 + C_3 + C_4)\gamma)^{3/2}\right\} \simeq \frac{M_\Phi^{24}}{m_\Phi^9}$$

$$\eta_t = \tau_t = \frac{1}{(c_0 + t)^{1/3}}$$

Here the bounds on $C$ and $\gamma$ are obtain after considering all 4 possible cases. We can also bound $\|w_1\| \le 2L_{g,0}M_\Phi$, thus $\|W_1\| \lesssim \frac{m_\Phi^3}{M_\Phi^7}$, and $\delta_{W,0} \le (\|W_1\| + \|W^\star(x_0)\|)^2 \lesssim \frac{1}{m_\Phi}$.

Substituting in equation 10 we finally obtain:

$$\frac{1}{2(T+1)}\mathbb{E}\left[\sum_{t=0}^{T}\|\nabla F_{\Phi^\epsilon}(x_t)\|^2\right] = \frac{1}{\eta_T T}\left[\frac{F_\Phi(x_0) - F_\Phi(x^\star)}{\gamma} + C\mathbb{E}[\delta_{W,0}]\right.$$

$$\left. + \frac{\mathbb{E}[\delta_{gy,0} + \delta_{fx,0} + \delta_{fy,0} + \delta_{gxy,0} + \delta_{gyy,0}]}{\gamma\eta_0} + \frac{\mathcal{O}(\ln(T+2))}{\gamma}\right]$$

$$\lesssim \frac{1}{\eta_T T}\left[\frac{M_\Phi^{12}}{m_\Phi^4} + \frac{M_\Phi^{18}}{m_\Phi^5}\frac{1}{m_\Phi} + \frac{M_\Phi^{26}}{m_\Phi^7} + \frac{M_\Phi^{12}}{m_\Phi^4}\mathcal{O}(\ln(T+2))\right]$$

In particular, following the analysis in Cutkosky & Orabona (2019), the sample complexity is $\tilde{\mathcal{O}}\left(\frac{1}{\epsilon^3}\frac{\text{Poly}(M_\Phi(\epsilon))}{\text{Poly}(m_\Phi(\epsilon))}\right)$.

# E    CHEBYSHEV SERIES: UNIFORM CONVERGENCE AND CONDITIONING

For completeness, we recall the definitions of $d$-dimensional Chebyshev polynomial and the Bernstein ellipse.

**Definition E.1.** Let $k = (k_1, ..., k_d) \in \mathbb{N}^d$ be a multi-index. The $k$-th $d$-dimensional Chebyshev polynomial is defined as:

$$\varphi : (\xi_1, ..., \xi_d) \in [-1, 1]^d \mapsto \prod_{j=1}^{d} T_{k_j}(\xi_j)$$

where $T_{k_j}(\xi_j) = \cos(k_j \arccos(\xi_j))$ is the Chebyshev polynomial of the first kind with degree $k_j$ in the $j$-th dimension.

**Definition E.2.** The Bernstein ellipse with parameter $\rho$ is the complex set defined as:

$$\left\{ z \in \mathbb{C} \ : \ z = \frac{e^{i\theta} + \rho e^{-i\theta}}{2}, \ \theta \in [0, 2\pi) \right\}$$

**Lemma E.3.**
*Under assumptions 3.1-(ii) and condition (c.3), $y^\star(x, \xi)$ is real-analytic over $\Xi$ for any fixed $x \in \mathbb{R}^{d_x}$.*

*Proof.* Recall that $G(x, y, \xi) \triangleq \mathbb{E}_{\eta \sim \mathbb{P}_{\eta|\xi}}[g(x, y, \xi, \eta)]$. Under assumption 3.1-(ii), $g$ is $\mu$-strongly convex in $y$ for any fixed $(x, \xi, \eta)$. Thus, $G$ is $\mu$-strongly convex in $y$ for any fixed $(x, \xi)$ and $y^\star(x, \xi)$ is the unique solution to $\nabla_2 G(x, y^{\cdot}\xi) = 0$. Additionally, since $G$ is real-analytic in $(y, \xi)$ for all $x \in \mathbb{R}^{d_x}$, so is $\nabla_2 G$. Define $H(x, y, \xi) = \nabla_2 G(x, y, \xi)$. Then $H(x, y^\star(x, \xi), \xi) = 0$ for all $x \in \mathbb{R}^{d_x}$. Further, the strong convexity of $G$ with respect to $y$ gives that $\nabla_2 H = \nabla_{22}^2 G$ is invertible. We can then use the analytic implicit function theorem to obtain that, for any fixed $x \in \mathbb{R}^{d_x}$, there exists a unique function $y^\dagger(x, \xi)$ real-analytic over $\Xi$ and solution to $H(x, y, \xi) = 0$ for all $\xi \in \Xi$. By unicity, we must have $y^\star = y^\dagger$ and it follows that $y^\star$ is real-analytic over $\Xi$ for any fixed $x \in \mathbb{R}^{d_x}$. $\qquad\square$

**Lemma E.4.**
*Let $f$ be an analytic function in $[-1, 1]^m$ that is analytically continuable to the open region $E_\rho$ delimited by the Bernstein ellipse with parameter $\rho \in (1, e^{1/2}]$, where it satisfies $|f(x)| \le M$ for all $x \in \mathcal{R}(E_\rho)$. Then for each $k \ge 0$ its Chebyshev coefficients satisfy*

$$|a_k| \le 2M\rho^{-k}.$$

*Proof.* Let $F : \begin{cases} E_\rho & \to & \mathbb{R} \\ z & \mapsto & f\left(\frac{z+z^{-1}}{2}\right) \end{cases}$ . Since $f$ is analytic in $\mathbb{E}_\rho$, $F$ is also analytic in $E_\rho$ as the composition of the two analytic functions $z \mapsto (z + z^{-1})/2$ and $x \mapsto f(x)$. From Theorem 3.1 of Trefethen (2019), the Chebyshev coefficients are given by

$$a_0 = \frac{2}{\pi i} \int_{|z|=1} z^{-1} F(z) dz$$

and

$$a_k = \frac{1}{\pi i} \int_{|z|=1} z^{-(1+k)} F(z) dz, \quad \forall k \ge 1.$$

If $F$ is analytic in the closure of $E_\rho$, we can expand the contour to $|z| = \rho$ without changing the value of these integrals. Since $|F(z)| \le M$ for all $z \in E_\rho$ and $\rho \in (1, e^{1/2}]$ we obtain for $k = 0$:

$$
\begin{aligned}
|a_0| &= \frac{2}{\pi} \left| \int_{|z|=\rho} z^{-1} F(z) dz \right| \\
&\le \frac{M}{\pi} \int_{|z|=\rho} |z|^{-1} dz \\
&= 4M \ln(\rho) \\
&\le 2M
\end{aligned}
$$

and similarly for all $k \ge 1$:

$$
\begin{aligned}
|a_k| &= \frac{1}{\pi} \left| \int_{|z|=\rho} z^{-(1+k)} F(z) dz \right| \\
&\le \frac{M}{\pi} \int_{|z|=\rho} |z|^{-(1+k)} dz \\
&= 2M\rho^{-k}
\end{aligned}
$$

Otherwise, we can expand the contour to $|z| = s$ for any $s < \rho$, giving the same bound for all $s < \rho$ and thus also for $s = \rho$.
Therefore $|a_k| \le 2M\rho^{-k}$ holds for any $k \ge 0$. $\qquad\square$

**Lemma E.5.**
*Let $f$ be an analytic function in $[-1, 1]^m$ that is analytically continuable to the open region $E_\rho^m$ delimited by $m$-dimensional Bernstein space $\prod_{i=1}^m E_\rho$ with $\rho \in (1, e^{1/2}]$ and $|f(x)| \leq M$ for all $x \in \mathcal{R}\left(E_\rho^m\right)$.*
*Then the coefficients of the $m$-dimensional Chebychev expansion:*

$$f(x) = \sum_{k_1=0}^{\infty} \cdots \sum_{k_m=0}^{\infty} a_{k_1,\ldots,k_m} \prod_{i=1}^m T_{k_i}(x_i)$$

*are such that, for any $k_1,\ldots,k_m \geq 0$,*

$$|a_{k_1,\ldots,k_m}| \leq M \prod_{i=1}^m \left(2\rho^{-k_i}\right).$$

*Proof.* Let $R_\rho^l \triangleq \mathcal{R}\left(E_\rho^l\right)$ be the projection of $E_\rho^l$ onto $\mathbb{R}^l$. We begin by defining two classes of functions:

$$H^{(l)}(M) = \left\{ f : [-1, 1]^l \to \mathbb{R} \mid f \text{ analytically continuable to } E_\rho^l, \quad \sup_{x \in R_\rho^l} |f(x)| \leq M \right\}$$

$$P^{(l)}(M) = \left\{ f : [-1, 1]^l \to \mathbb{R} \mid \text{the Chebychev coefficients of } f \text{ satisfy } |a_{k_1,\ldots,k_l}| \leq M \prod_{i=1}^l \left(2\rho^{-k_i}\right) \right\}$$

Let the hypothesis of induction be that the statement holds for a dimension $l - 1 \leq m$. Namely, $H^{(l-1)}(M) \subseteq P^{(l-1)}(M)$. We want to show that $H^{(l)}(M) \subseteq P^{(l)}(M)$.

Let $f^{(l)} \in H^{(l)}(M)$. For any fixed $(x_1,\ldots,x_{l-1}) \in [-1, 1]^{l-1}$, define the single-variable function:

$$f_l^{(l)}(x_1,\ldots,x_{l-1})[x_l] \triangleq f^{(l)}(x_1,\ldots,x_{l-1},x_l), \quad \forall x_l \in [-1, 1].$$

Note that since $f^{(l)} \in H^{(l)}$ we have in particular for any fixed $(x_1,\ldots,x_{l-1}) \in R_\rho^{l-1}$ that $\sup_{x_l \in R_\rho} \left| f_l^{(l)}(x_1,\ldots,x_{l-1})[x_l] \right| \leq \sup_{x \in R_\rho^l} \left| f^{(l)}(x) \right| \leq M$. Since $f^{(l)}$ is jointly real-analytic in all its variables and can be continued analytically to $E_\rho^l$, the mapping $x_l \mapsto f_l^{(l)}(z_1,\ldots,z_{l-1})[x_l]$ is also real-analytic and can be continued analytically to $E_\rho$ for any $(z_1,\ldots,z_{l-1}) \in E_\rho^{l-1}$.

Applying Lemma E.4, we have that the coefficients of the Chebyshev expansion:

$$f_l^{(l)}(x_1,\ldots,x_{l-1})[x_l] = \sum_{k_l=1}^{\infty} c_{k_l}(x_1,\ldots,x_{l-1}) T_{k_l}(x_l).$$

satisfy

$$|c_{k_l}(x_1,\ldots,x_{l-1})| \leq 2M\rho^{-k_l}, \quad \forall k_l \geq 0.$$

Since this hold for any fixed $(x_1,\ldots,x_{l-1}) \in R_\rho^{l-1}$, we have $\sup_{x \in R_\rho^{l-1}} |c_{k_l}(x)| \leq 2M\rho^{-k_l}$.

Further, because $f^{(l)}$ is jointly analytic in all its variables and can be continued analytically to $E_\rho^l$, for any fixed $x_l$, the mapping $(x_1,\ldots,x_{l-1}) \mapsto f^{(l)}(x_1,\ldots,x_{l-1},x_l)$ is also analytic and can be continued analytically to $E_\rho^{l-1}$. By definition,

$$c_{k_l}(x_1,\ldots,x_{l-1}) \triangleq \frac{2}{\pi} \int_0^\pi f_l^{(l)}(x_1,\ldots,x_{l-1})[\cos(\theta)] T_{k_l}(\cos\theta) d\theta$$

$$= \frac{2}{\pi} \int_0^\pi f^{(l)}(x_1,\ldots,x_{l-1},\cos(\theta)) T_{k_l}(\cos\theta) d\theta$$

so $c_{k_l}$ is analytic on $[-1,1]^{l-1}$ and can be continued analytically on $E_\rho^{l-1}$ as integration preserves analyticity. Therefore, $c_{k_l} \in H^{(l-1)}(2M\rho^{-k_l})$. By the induction hypothesis, it follows that $c_{k_l} \in P^{(l-1)}(2M\rho^{-k_l})$. Thus, each $c_{k_l}$ can be expanded as:

$$c_{k_l}(x_1, \ldots, x_{l-1}) = \sum_{k_1=1}^{\infty} \cdots \sum_{k_{l-1}=1}^{\infty} a_{k_1,\ldots,k_{l-1},k_l} \prod_{i=1}^{l-1} T_{k_i}(x_i)$$

with

$$|a_{k_1,\ldots,k_{l-1},k_l}| \le \left(2M\rho^{-k_l}\right) \prod_{i=1}^{l-1} \left(2\rho^{-k_i}\right)$$

$$= M \prod_{i=1}^{l} \left(2\rho^{-k_i}\right)$$

It follows that for any $(x_1, \ldots, x_l) \in [-1,1]^l$,

$$f^{(l)}(x_1, \ldots, x_l) = f_l^{(l)}(x_1, \ldots, x_{l-1})[x_l]$$

$$= \sum_{k_l=0}^{\infty} c_{k_l}(x_1, \ldots, x_{l-1}) T_{k_l}(x_l)$$

$$= \sum_{k_l=0}^{\infty} \left[ \sum_{k_1=0}^{\infty} \cdots \sum_{k_{l-1}=0}^{\infty} a_{k_1,\ldots,k_{l-1},k_l} \prod_{i=1}^{l-1} T_{k_i}(x_i) \right] T_{k_l}(x_l)$$

$$= \sum_{k_1=0}^{\infty} \cdots \sum_{k_l=0}^{\infty} a_{k_1,\ldots,k_{l-1},k_l} \prod_{i=1}^{l} T_{k_i}(x_i)$$

with $|a_{k_1,\ldots,k_{l-1},k_l}| \le M \prod_{i=1}^{l} \left(2\rho^{-k_i}\right)$. This shows that $f^{(l)} \in P^{(l)}(M)$, completing the induction step.

The initialization ($l = 1$) of the induction reduces to Lemma E.4. We conclude by induction that:

$$H^{(m)}(M) \subseteq P^{(m)}(M)$$

□

**Lemma E.6.**
*Let $f = \lim_{n\to\infty} f_n$ where*

$$f_n : \begin{cases} [-1,1]^m & \to & \mathbb{R} \\ x & \mapsto & \sum_{k_1=0}^{n} \cdots \sum_{k_m=0}^{n} a_{k_1,\ldots,k_m} \prod_{i=1}^{m} T_{k_i}(x_i) \end{cases}, \quad \forall n \ge 1$$

*and $a$ satisfies for $\rho > 1$:*

$$|a_{k_1,\ldots,k_m}| \le M \prod_{i=1}^{m} \left(2\rho^{-k_i}\right) \quad \forall k_1, \ldots, k_m \ge 0.$$

*Then the residual $r_n = \sup_{x \in [-1,1]^m} |f(x) - f_n(x)|$ is bounded by*

$$r_n \le M \left(\frac{2}{\rho - 1}\right)^m \left[1 - (1 - (1/\rho)^n)^m\right].$$

*Proof.* By definition, the remainder after truncation is

$$f(x) - f_n(x) = \sum_{\substack{k_1,\ldots,k_m \ge 0 \\ \exists j \text{ s.t. } k_j > n}} a_{k_1,\ldots,k_m} \prod_{i=1}^{m} T_{k_i}(x_i).$$

Since $|T_{k_i}(x_i)| \leq 1$ for all $x_i \in [-1, 1]$, we have

$$r_n \leq \sup_{\substack{x \in [-1,1]^m}} \sum_{\substack{k_1, \ldots, k_m \geq 0 \\ \exists j \text{ s.t. } k_j > n}} |a_{k_1, \ldots, k_m}| \prod_{i=1}^m |T_{k_i}(x_i)|$$

$$\leq \sum_{\substack{k_1, \ldots, k_m \geq 0 \\ \exists j \text{ s.t. } k_j > n}} |a_{k_1, \ldots, k_m}|.$$

Since $|a_{k_1, \ldots, k_m}| \leq M \prod_{i=1}^m (2\rho^{-k_i})$, it follows that

$$r_n \leq M \sum_{\substack{k_1, \ldots, k_m \geq 0 \\ \exists j \text{ s.t. } k_j > n}} \prod_{i=1}^m (2\rho^{-k_i})$$

$$= M \sum_{k_1=0}^{\infty} \cdots \sum_{k_m=0}^{\infty} \prod_{i=1}^m (2\rho^{-k_i}) - M \sum_{k_1=0}^{n} \cdots \sum_{k_m=0}^{n} \prod_{i=1}^m (2\rho^{-k_i})$$

$$= M \left( \sum_{k=0}^{\infty} 2\rho^{-k} \right)^m - M \left( \sum_{k=0}^{n} 2\rho^{-k} \right)^m$$

Since $\rho > 1$ we have:

$$\sum_{k=0}^{\infty} 2\rho^{-k} = \frac{2}{\rho - 1}$$

for the full sum, and

$$\sum_{k=0}^{n} 2\rho^{-k} = 2 \cdot \frac{1 - (1/\rho)^n}{\rho - 1}$$

for the truncated sum.

Substituting this back into our bound, we get:

$$r_n \leq M \left[ \left( \frac{2}{\rho - 1} \right)^m - \left( \frac{2(1 - (1/\rho)^n)}{\rho - 1} \right)^m \right]$$

$$= M \left( \frac{2}{\rho - 1} \right)^m [1 - (1 - (1/\rho)^n)^m]$$

which gives the desired result. $\qquad \square$

**Proposition E.7.** *Let $\Phi$ be the basis of $d_\xi$-dimensional Chebyshev polynomials and*

$$\underline{N}(\tilde{\epsilon}) = \mathcal{O}\left( \ln^{d_\xi}\left( \tilde{\epsilon}^{-1} \right) \right).$$

*If assumption 3.1 and the conditions of Theorem 4.5 hold, then $N_\Phi$ is expressive with $N_\Phi(\epsilon) = \underline{N}\left( \frac{\epsilon}{2K} \sqrt{\frac{\mu}{L_{g,1}}} \right)$.*

*Proof.* Since $\Xi$ is bounded under condition (c.2), we assume without loss of generality that $\Xi = [-1, 1]^{d_\xi}$. Indeed, the domain can be normalized by defining a scaling mapping $S : \Xi \to [-1, 1]^{d_\xi}$ and replacing $\xi$ with $S(\xi)$ in (CSBO) and (SBO$_{\Phi^\epsilon}$). Under assumptions 3.1-(ii) and condition (c.3), we have from Lemma E.3 that $y^\star(x, \cdot)$ is real-analytic over $\Xi$ for any fixed $x \in \mathbb{R}^{d_x}$. Hence there exists $\rho \in (1, e^{1/2})$ such that $y^\star(x, \cdot)$ is analytically continuable to the closure of $E_\rho$, the open region delimited by the $d_\xi$-dimensional Bernstein space $\prod_{i=1}^m E_\rho$. Since $y^\star(x, \cdot)$ is analytic on the compact

set $\overline{E_\rho}$, it is in particular continuous and $y^\star(x, \cdot)$ is bounded on $E_\rho$ by some constant $M$. For any $j \in [d_y]$ Using Lemma E.5, $y_j^\star(x, \cdot)$ admits the $d_\xi$-dimensional Chebyshev expansion:

$$y_j^\star(x, \xi) = \sum_{k_1=0}^{\infty} \cdots \sum_{k_{d_\xi}=0}^{\infty} a_{k_1,\ldots,k_{d_\xi}}^{(j)} \prod_{i=1}^{d_\xi} T_{k_i}(\xi_i)$$

where for any $k_1, \ldots, k_m \geq 0$,

$$|a_{k_1,\ldots,k_{d_\xi}}^{(j)}| \leq M \prod_{i=1}^{d_\xi} \left(2\rho^{-k_i}\right).$$

Let $W^\dagger$ be such that its $j$-th row contains the elements of $a^{(j)}$. Then

$$y_{\Phi,j}(W^\dagger(x), \xi) = W_j^\dagger(x)\Phi(\xi)$$

$$= \sum_{k_1=0}^{n} \cdots \sum_{k_{d_\xi}=0}^{n} a_{k_1,\ldots,k_{d_\xi}} \prod_{i=1}^{d_\xi} T_{k_i}(\xi_i)$$

Lemma E.6 then yields for any $n \geq 1$:

$$\sup_{\xi \in \Xi} \left|y_{\Phi,j}(W^\dagger(x), \xi) - y_j^\star(x, \xi)\right| \leq M \left(\frac{2}{\rho - 1}\right)^{d_\xi} \left[1 - (1 - (1/\rho)^n)^{d_\xi}\right] \qquad (11)$$

Define:

$$\underline{n}(\tilde{\epsilon}) \triangleq \left\lceil -\ln\left(1 - \left(1 - \frac{\tilde{\epsilon}}{M\sqrt{d_y}}\left(\frac{\rho - 1}{2}\right)^{d_\xi}\right)^{1/d_\xi}\right) / \ln(\rho)\right\rceil \quad \text{and} \quad \underline{N}(\tilde{\epsilon}) \triangleq \underline{n}(\tilde{\epsilon})^{d_\xi} \quad (12)$$

Note that $\underline{N}(\tilde{\epsilon}) = \Theta\left(\ln^{d_\xi}(1/\tilde{\epsilon})\right)$ as $\tilde{\epsilon} \to 0$. Furthermore, for any number of basis functions $N \geq \underline{N}(\tilde{\epsilon})$, we have at least $n = N^{1/d_\xi} \geq \underline{n}(\tilde{\epsilon})$ elements per dimension. As the right hand side of equation 11 decreases in $n$, we have for any $n \geq \underline{n}(\tilde{\epsilon})$:

$$\sup_{\xi \in \Xi} \left|y_{\Phi,j}(W^\dagger(x), \xi) - y_j^\star(x, \xi)\right| \leq \frac{\tilde{\epsilon}}{\sqrt{d_y}}, \quad \forall j \in [d_y]$$

Since this holds for any $j \in [d_y]$, we obtain:

$$\sup_{\xi \in \Xi} \left\|y_\Phi(W^\dagger(x), \xi) - y^\star(x, \xi)\right\|^2 \leq \tilde{\epsilon}^2.$$

$\square$

**Lemma E.8.** *Let $A^{(N)} \in \mathbb{R}^{N \times N}$ be a zero-indexed matrix containing the unweighted scalar product of $d$-dimensional Chebyshev polynomials. Then $\lambda_{min}\left(A^{(N)}\right) = \Omega\left(N^{-1}\right)$.*

*Proof.* We first consider the 1-dimensional case and define $B \in \mathbb{R}^{n \times n}$ satisfying:

$$B_{i,j} = \frac{1}{2}\int_{-1}^{1} T_i(x)T_j(x)dx$$

For any zero-indexed vector $v \in \mathbb{R}^n$, we have:

$$v^\top B v = \sum_{i=0}^{n-1}\sum_{j=0}^{n-1} \frac{1}{2}v_i v_j \int_{-1}^{1} T_i(x)T_j(x)dx$$

$$= \frac{1}{2}\int_{-1}^{1}\left(\sum_{i=0}^{n-1} v_i T_i(x)\right)^2 dx$$

$$= \frac{1}{2}\int_{-1}^{1}\left(\sum_{i=0}^{n-1} v_i \cos(i\arccos(x))\right)^2 dx$$

After substituting $x = \cos(\theta)$ and $dx = -sin(\theta)d\theta$ we obtain for $\delta = \frac{1}{4n}$:

$$v^\top B v = \frac{1}{2} \int_0^\pi \left( \sum_{i=0}^{n-1} v_i \cos(i\theta) \right)^2 \sin(\theta) d\theta$$

$$\geq \frac{1}{2} \int_\delta^{\pi-\delta} \left( \sum_{i=0}^{n-1} v_i \cos(i\theta) \right)^2 \sin(\theta) d\theta$$

$$\geq \frac{\sin(\delta)}{2} \int_\delta^{\pi-\delta} \left( \sum_{i=0}^{n-1} v_i \cos(i\theta) \right)^2 d\theta$$

On one hand, we know from Fourier theory that

$$\int_0^\pi \cos(i\theta) \cos(j\theta) = \begin{cases} 0 & \text{if } i \neq j \\ \pi & \text{if } i = j = 0 \\ \frac{\pi}{2} & \text{otherwise.} \end{cases}$$

and thus

$$\int_0^\pi \left( \sum_{i=0}^{n-1} v_i \cos(i\theta) \right)^2 d\theta = \pi v_0^2 + \frac{\pi}{2} \sum_{i=1}^{n-1} v_i^2$$

$$\geq \frac{\pi}{2} \|v\|^2$$

On the other hand, we have using Cauchy-Schwartz inequality:

$$\int_0^\pi \left( \sum_{i=0}^{n-1} v_i \cos(i\theta) \right)^2 d\theta = \int_\delta^{\pi-\delta} \left( \sum_{i=0}^{n-1} v_i \cos(i\theta) \right)^2 d\theta + \int_{[0,\delta] \cup [\pi-\delta,\pi]} \left( \sum_{i=0}^{n-1} v_i \cos(i\theta) \right)^2 d\theta$$

$$\leq \int_\delta^{\pi-\delta} \left( \sum_{i=0}^{n-1} v_i \cos(i\theta) \right)^2 d\theta + \|v\|^2 \int_{[0,\delta] \cup [\pi-\delta,\pi]} \left( \sum_{i=0}^{n-1} \cos^2(i\theta) \right) d\theta$$

$$\leq \int_\delta^{\pi-\delta} \left( \sum_{i=0}^{n-1} v_i \cos(i\theta) \right)^2 d\theta + 2\delta \|v\|^2 n$$

Hence we obtain:

$$\int_\delta^{\pi-\delta} \left( \sum_{i=0}^{n} v_i \cos(i\theta) \right)^2 d\theta \geq \left( \frac{\pi}{2} - 2\delta n \right) \|v\|^2$$

and we conclude using $\sin(\delta) \geq \frac{\delta}{2}$ for any $\delta \in (0,1)$ that:

$$v^\top B v \geq \frac{\sin(\delta)}{2} \int_\delta^{\pi-\delta} \left( \sum_{i=0}^{n} v_i \cos(i\theta) \right)^2 d\theta$$

$$\geq \frac{\delta}{4} \left( \frac{\pi}{2} - 2\delta n \right) \|v\|^2$$

$$= \frac{1}{16n} \left( \frac{\pi}{2} - \frac{1}{2} \right) \|v\|^2$$

Therefore we obtain for all $n \in \mathbb{N}^*$ that $\lambda_{\min}(B) \geq \frac{c_B}{n} I_n$ with $c_B = \frac{\pi-1}{32}$.

Suppose first that $N = n^d$ for some $n \in \mathbb{N}^*$. We can decompose $A^{(N)} = \bigotimes_{i=1}^{d} B$. Using Lemma C.1 we obtain:

$$A^{(N)} \succeq \left( \frac{c_B}{n} \right)^d I_N$$

$$= \frac{c_B^d}{N} I_N$$

Consider now an arbitrary $N \in \mathbb{N}^*$, and $n \in \mathbb{N}^*$ such that $N \in \left[(n-1)^d + 1, n^d\right]$. Since $A^{(N)}$ is a principal submatrix of $A^{(n^d)}$, the Eigenvalue Interlacing Theorem gives:

$$\lambda_{\min}\left(A^{(N)}\right) \geq \lambda_{\min}\left(A^{(n^d)}\right)$$
$$\geq \frac{c_B^d}{n^d}$$
$$= \frac{c_B^d}{\lceil N^{1/d}\rceil^d}$$

where the second inequality uses the result of the case $N = n^d$, and the last equality hold since $N \in \left[(n-1)^d + 1, n^d\right]$.

We conclude that $\lambda_{\min}\left(A^{(N)}\right) = \Omega\left(N^{-1}\right)$.

$\square$

**Proposition E.9.** *Let $\Phi$ be the basis of $d_\xi$-dimensional Chebyshev polynomials. If condition (c.1) holds, then $m_\Phi(\epsilon) = \Omega\left(\frac{1}{N_\Phi(\epsilon)}\right)$.*

*Proof.* Under condition (c.1), we have that $\Xi$ is finite or there exists $\underline{c} > 0$ such that the density of $\mathbb{P}_\xi$ is lower bounded by $\underline{c}$ on $\Xi$.

Suppose first that $|\Xi|$ is finite. Then for any $N \leq |\Xi|$ we have that $\{\Phi(\xi)\}_{\xi \in \Xi}$ spans $\mathbb{R}^N$. Thus there exists $c_N > 0$ such that $\Sigma_\Phi \succeq c_N I_N$. Since $n-1$ polynomials of distinct order can interpolate $n$ point, we have $N_\Phi(\epsilon) \leq |\Xi|$. Therefore, for any $N \in \mathbb{N}^*$ we have $\Sigma_\Phi \succeq \min_{n=1}^{|\Xi|} c_n I_N$ and in particular $m_\Phi(\epsilon) \geq c \geq \frac{c}{N_\Phi(\epsilon)}$ with $c = \min_{n=1}^{|\Xi|} c_n$.

Suppose now that the density of $\mathbb{P}_\xi$ is lower bounded by $\underline{c}$ on $\Xi$. Then:

$$\Sigma_\Phi = \mathbb{E}_\xi\left[\Phi(\xi)\Phi(\xi)^\top\right]$$
$$= \underline{c}\int_{-1}^1 \Phi(\xi)\Phi(\xi)^\top d\xi + \int_{-1}^1 \Phi(\xi)\Phi(\xi)^\top(d\mathbb{P}(\xi) - \underline{c}d_\xi)$$

From Lemma E.8 we have $\lambda_{\min}\left(\int_{-1}^1 \Phi(\xi)\Phi(\xi)^\top d\xi\right) = \Omega(1/N_\Phi(\epsilon))$. Additionally, $(d\mathbb{P}(\xi) - \underline{c}d_\xi)\Phi(\xi)\Phi(\xi)^\top \succeq 0$ for any $\xi \in \Xi$ as the product of a positive term and a rank 1 matrix. Therefore we have:

$$\lambda_{\min}\left(\Sigma_\Phi\right) = \Omega\left(1/N_\Phi(\epsilon)\right)$$

and we conclude that $m_\Phi(\epsilon) = \Omega(1/N_\Phi(\epsilon))$. $\square$

Combining the above results, we now prove Theorem 4.5.

*Proof.* Suppose that assumption 3.1 and the conditions of Theorem 4.5 hold. The first statement of the theorem follows from Proposition E.7. Indeed we have that $\Phi$ is expressive with $N_\Phi(\epsilon) = \underline{N}\left(\frac{\epsilon}{2K}\sqrt{\frac{\mu}{L_{g,0}}}\right)$ and since $\underline{N}(\tilde{\epsilon}) = O\left(\ln^{d_\xi}(\tilde{\epsilon}^{-1})\right)$, we obtain $N_\Phi(\epsilon) = O\left(\ln^{d_\xi}(\epsilon^{-1})\right)$. Since $\Phi$ encodes multivariate Chebyshev polynomials, we have $|\Phi_i| \leq 1$ for any $i \in [N_\Phi(\epsilon)]$ and thus $M_\Phi(\epsilon) \leq \sqrt{N_\Phi(\epsilon)} = O\left(\ln^{d_\xi/2}(\epsilon^{-1})\right)$. The second statement directly follows from Proposition E.9, where substituting $N_\Phi(\epsilon) = O\left(\ln^{d_\xi}(\epsilon^{-1})\right)$ into $m_\Phi(\epsilon) = \Omega\left(\frac{1}{N_\Phi(\epsilon)}\right)$ gives $m_\Phi(\epsilon) = \Omega\left(\ln^{-d_\xi}(\epsilon^{-1})\right)$. $\square$

## F  DISCLOSURE OF LARGE LANGUAGE MODELS USAGE

LLMs were used solely to improve the clarity and readability of the main text (e.g., grammar and phrasing). No technical content, proofs, or experimental results were generated or modified using LLMs.

