# OpenReview forum: "Reducing Contextual Stochastic Bilevel Optimization via Structured Function Approximation"
_ICLR.cc/2026/Conference — ICLR 2026 Poster_

### Official Review · Reviewer_YLFf · 2025-10-26

**Soundness:** 2
**Presentation:** 3
**Contribution:** 2
**Rating:** 2
**Confidence:** 5

**Summary:**

This paper reduces the contextual stochastic bilevel optimization (CSBO) to standard stochastic bilevel optimization (SBO) via a function approximation characterized as a linear combination of basis functions. Under certain conditions, the CSBO and SBO have the same complexity of finding small hypergradient. Numerical experiments on inverse optimization and hyperparameter optimization seem to show the effectiveness of the proposed algorithm.

**Strengths:**

1. The idea of reduction from CSBO to SBO is novel. To the best of my knowledge, this aspect is new in the literature.
2. The presentation of the paper is clear.

**Weaknesses:**

1. The conditions required in Theorem 4.5 (c.1, c.2, c.3) are too strong. For example, as the authors described in line 68-line 69, the approach of Guo et al. 2021 is not amenable to settings where $\xi$ is continuous. However the condition c.1 also requires $\xi$ is discrete with finite cardinality. Therefore, it is unclear how this paper improves over Guo et al. 2021. Since these conditions are crucial to keep the complexity of CSBO the same as the SBO, this is a significant weakness. I suggest the authors to include a table to have a comprehensive comparison of this paper versus other baselines.

2. It is unclear the relevance of the results in machine learning community. Are there any practical machine learning examples satisfying conditions (c.1, c.2, c.3), and using this paper's algorithm can provably improve state-of-the-art bilevel optimization baselines? In addition, what if these conditions do not hold? It may significantly increase the complexity of the algorithm.

3. The experimental results are very weak. It is only tested on toy problems at small scale (e.g., MNIST dataset). In addition, the only baseline the paper considered was stocBiO, but there are lots of other bilevel optimization baselines which are missed.

4. The formulation used in the task of hyperparameter optimization (Section 5.2) is not necessarily relevant in machine learning. For example, there are many formulations for the hyperparameter optimization tasks in the literature which use SBO instead of CSBO. Why the CSBO formula (5) is better?

5. Using fixed basis function is not necessarily good. The authors do not consider learned basis functions, which may yield stronger approximation guarantees.

**Questions:**

See the weakness section.

---

> ### Author Response · Authors · 2025-11-21
>
> Thank you for your detailed feedback. We address your concerns, particularly clarifying important misunderstandings about our contributions.
>
> **1. Critical clarification of condition c.1**
>
> Condition $c.1$ states: ``The cardinality of $\Xi$ is finite *OR* there exists $c > 0$ such that the density of $\mathbb{P}_\xi$ is lower bounded by $c$ on $\Xi$".
>
> This OR condition covers two classes of problems:
> - Finite/discrete context distribution encompassing SBO with multiple lower-level problems (generalizing Guo et al. [6])
> - Continuous contexts distribution with non-vanishing density. This covers most practical settings: uniform distribution, truncated Gaussians and Beta distributions, mixtures...
>
> The second class of problems goes strictly beyond the setting of Guo et al. [6]. The table below clarifies how our method relates to existing work.
>
> | | Sampling Oracle | SBO with $m$ subproblems | CSBO | CSBO + condition c.1-3
> |---|---|---|---|---|
> | Guo et al. [6] | Conditional | $O(m\epsilon^{-3})$ | X | X |
> | Hu et al. [1] | Conditional | $\tilde{O}(\epsilon^{-4})$ | $\tilde{O}(\epsilon^{-4})$ | $\tilde{O}(\epsilon^{-4})$ |
> | Ours | Joint only | $\tilde{O}(\epsilon^{-3})$ | $\tilde{O}\left(\epsilon^{-3} \cdot \text{poly}\left(M_\Phi(\epsilon), \frac{1}{m_\Phi(\epsilon)}\right)\right)$ | $\tilde{O}(\epsilon^{-3})$ |
>
> **2. Practical ML relevance and violated assumptions**
>
> The conditions c.1-3 are sufficient (not necessary) conditions for the specific choice of Chebyshev basis to yield near-optimal sampling guarantees. That being said, many ML applications satisfy our conditions:
> - Meta-learning [1]
> - Wasserstein DRO with Side Information [1]
> - Tax Design, Reward Shaping, and Dynamic Mechanism Design [2], which are naturally modeled with discrete or trucated continuous context distributions
>
> Regarding the performance of our approach in settings where our conditions do not hold, please see the additional experiment \#1 in the comment below, reporting the performance of the parametrization when the lower-level is nonconvex.
>
> **3. Experimental Validation**
>
> MNIST is the most prevalent benchmark in SBO, notably used in [3,4,5]. Despite the simplicity of MNIST as a lower-level, the data-cleaning task is non-trivial due to the dimension of the upper level ($d_x = N_\text{train} = 19,000$). We tested additional SBO backbones (SOBA, SUSTAIN), showing consistent improvements (see additional experiments \#2 in the comments below).
>
> **4. Relevance of CSBO**
>
> Because existing CSBO algorithms either require conditional sampling or incur suboptimal complexity, prior works replace the continuous context distribution with a few scenarios, optimizing a surrogate with no accuracy guarantee for the original problem, and resulting in solutions that can be arbitrarily biased by the discretization. We tackle the original problem with exact context distribution and without discretization bias. Generally, the relevance of CSBO in ML applications is developed in [1].
>
> **5. Learned and fixed bases**
>
> We agree that learned bases are promising in practice and list this as future work. However, data-dependent bases make theoretical guarantees challenging, as ensuring good conditioning (and thus preserving strong convexity of the reformulation) becomes difficult. While learned bases have the potential to deliver stronger empirical performance, they can only yield weaker theoretical guarantees.
>
> [1] "Contextual stochastic bilevel optimization", Hu et al.
>
> [2] "Stochastic Bilevel Optimization with Lower-Level Contextual Markov Decision Processes'' Thoma et al.
>
> [3] "A Fully First-Order Method for Stochastic Bilevel Optimization'', Kwon et al.
>
> [4] "Bilevel Optimization: Convergence Analysis and Enhanced Design'', Ji et al.
>
> [5] "A framework for bilevel optimization that enables stochastic and global variance reduction algorithms'', Dagréou et al.
>
> [6] "Randomized stochastic variance-reduced methods for multi-task stochastic bilevel optimization'', Guo et al.

---

> > ### Author Response · Authors · 2025-11-21
> > **Additional experiments #1: Scalability and non-convex lower-level**
> >
> > This experiment addresses the reviewers' concerns about 1) the scalability of our approach and 2) its applicability to problems violating our assumptions. This new, larger-scale experiment has a similar setting as the data-cleaning experiment presented in section 5.2, but involves a more complex task (CIFAR-100) and a 4-layer deep neural network instead of a linear classifier, resulting in non-convex lower and upper level objectives.
> >
> > Specifications: The dimensions of $x$ and $y$ are respectively 45k and 288k. The loss function is identical to the one given in Appendix A.1, except that the linear classifier is replaced with a frozen ResNet-18 combined with a 4-layer feed-forward network whose weights are given by $y$. Specifically, we follow [10]’s preprocessing by extracting a fixed feature vector of dimension 512 from each image using a pretrained ResNet-18 without its final layer. We then pass the feature vector to a 4‑layer feed‑forward network whose weights are determined by $y$, the decision variables of the lower level. The dataset consists of 45,000 images for training and 5,000 for validation. Images are evenly distributed across 100 distinct classes. The corruption level is maintained at $p=0.3$.
> >
> > Important note: this experiment lies outside the scope of our theoretical framework, as neither objective is convex in $y$, and the guarantees of strongly convex SBO no longer apply.
> >
> > Below, we report the final training and validation losses for $\texttt{stocBiO}$ with $n = 5, 10, 20, 50$, as well as for $R_\text{monomial}$ and $R_\text{Chebyshev}$ with $N = 2, 5, 50$, averaged over 5 randomly seeded runs.
> >
> > | Method   | n  | Training Loss | Validation Loss |
> > |----------|----|------------|----------|
> > | **stocBiO** | 5  | 2.011 $\pm$ 0.027        | 1.872 $\pm$ 0.037     |
> > |          | 10 | 1.872 $\pm$ 0.019       | 1.793 $\pm$ 0.032     |
> > |          | 20 | 1.836 $\pm$ 0.018       | 1.772 $\pm$ 0.025     |
> > |          | 50 | 2.090* $\pm$ 0.023       | 1.828* $\pm$ 0.031     |
> > | $\mathcal{R}_\text{monomial}$ | 2  | 1.773 $\pm$ 0.014       | 1.673 $\pm$ 0.029     |
> > |          | 5  | 1.743 $\pm$ 0.008       | 1.664 $\pm$ 0.034     |
> > |          | 50 | 1.750 $\pm$ 0.009       | 1.660 $\pm$ 0.029     |
> > | $\mathcal{R}_\text{Chebyshev}$ | 2  | 1.776 $\pm$ 0.023       | 1.672 $\pm$ 0.022     |
> > |          | 5  | 1.760 $\pm$ 0.021       | 1.660 $\pm$ 0.034     |
> > |          | 50 | 1.784 $\pm$ 0.028       | 1.669 $\pm$ 0.025     |
> >
> > *: $\texttt{stocBiO}[50]$ did not converge within the time limit.
> >
> > Similar to Figure 1 of the main text, the parametrization again achieves lower final training and validation losses in this setting. However, in contrast to the experiments presented in the main text, we observe that $R_\text{Chebyshev}$ and $R_\text{monomial}$ perform almost identically. Interestingly, despite the strong nonlinearity of the problem, near-optimal solutions appear to concentrate on a low-dimensional subspace, as bases with $N=2$ functions already match the performance of larger $N$. These findings suggest that our reduction method remains effective in large-scale, non-convex settings, as the gap in performance and memory efficiency over baselines is wider than that observed in the experiments in strongly convex settings.

---

> > ### Author Response · Authors · 2025-11-21
> > **Additional Experiment #2: Additional solvers**
> >
> > The experimental setup follows the same data-cleaning task described in section 5.2. To demonstrate that our findings are not tied to a particular choice of backbone solver, we expanded the evaluation to include two additional bilevel optimization solvers which differ substantially in design from $\texttt{stocBiO}$: $\texttt{SUSTAIN}$ (Khanduri et al, 2021) and $\texttt{SOBA}$ (Dagréou et al., 2022). We compare the final training and validation losses of the solutions obtained on the $n$-subproblems SBO approximation ($\texttt{solver}$) with the solutions obtained on the single lower-level reformulation using $n$ Chebyshev basis functions ($\mathcal{R}_\text{Chebyshev}[\texttt{solver}]$). The results are averaged over 5 randomly seeded runs.
> >
> > As shown in the table below, for any fixed $n$, the parametrization consistently improves both training and validation losses across all solvers. Further, a lower final loss can be achieved with fewer basis functions than subproblems. This indicates that the advantages of our parametrization are not specific to $\texttt{stocBiO}$, but persist across several distinct bilevel solvers.
> >
> > | Method   | n  | Validation Loss | Training Loss |
> > |----------|----|-------------|-------------------|
> > | $\texttt{SOBA}$ | 5 | 0.869 $\pm$ 0.007 |1.208 $\pm$ 0.021 |
> > |  | 10 | 1.015 $\pm$ 0.011 |1.405 $\pm$ 0.017 |
> > |  | 50 | 1.348 $\pm$ 0.024 |1.838 $\pm$ 0.034 |
> > | $\mathcal{R}_\text{Chebyshev}[\texttt{SOBA}]$ | 5 | 0.659 $\pm$ 0.014 |0.790 $\pm$ 0.030 |
> > |  | 10 | 0.653 $\pm$ 0.013 |0.783 $\pm$ 0.027 |
> > |  | 50 | 0.656 $\pm$ 0.012 |0.786 $\pm$ 0.029 |
> > | | | | |
> > | $\texttt{SUSTAIN}$ | 5 | 1.133 $\pm$ 0.006 |1.335 $\pm$ 0.025 |
> > |  | 10 | 1.167 $\pm$ 0.010 |1.540 $\pm$ 0.015 |
> > |  | 50 | 1.326 $\pm$ 0.018 |1.887 $\pm$ 0.018 |
> > | $\mathcal{R}_\text{Chebyshev}[\texttt{SUSTAIN}]$ | 5 | 0.890 $\pm$ 0.011 |0.972 $\pm$ 0.026 |
> > |  | 10 | 0.905 $\pm$ 0.008 |0.975 $\pm$ 0.012 |
> > |  | 50 | 0.903 $\pm$ 0.009 |0.968 $\pm$ 0.028 |
> > | | | | |
> > | $\texttt{stocBio}$ | 5 | 0.941 $\pm$ 0.010 |0.848 $\pm$ 0.027 |
> > |  | 10 | 0.837 $\pm$ 0.020 |0.784 $\pm$ 0.036 |
> > |  | 50 | 0.734 $\pm$ 0.022 |0.739 $\pm$ 0.033 |
> > | $\mathcal{R}_\text{Chebyshev}[\texttt{stocBio}]$ | 5 | 0.715 $\pm$ 0.022 |0.740 $\pm$ 0.041 |
> > |  | 10 | 0.706 $\pm$ 0.010 |0.729 $\pm$ 0.023 |
> > |  | 50 | 0.713 $\pm$ 0.017 |0.733 $\pm$ 0.026 |
> > | | | | |
> >
> > Note that all runs use the same outer-iteration budget of $3 \cdot 10^4$. While $\texttt{stocBiO}$ can efficiently handle many subproblems with a large inner-loop budget $T_{\text{inner}}$, single-loop methods perform poorly in this regime. This is because, in single-loop algorithms, each outer iteration updates only one of the $n$ lower-level variables, so many subproblems remain stale when $n$ is large. Our parametrized reformulation instead updates a single shared lower-level parameter at every step, producing smoother context-dependent updates and therefore scales better with $n$.

---

### Official Review · Reviewer_yLsY · 2025-10-31

**Soundness:** 3
**Presentation:** 3
**Contribution:** 3
**Rating:** 6
**Confidence:** 3

**Summary:**

The paper focuses on Contextual Stochastic Bilevel Optimization (CSBO) and proposes a reduction framework that approximates lower-level solutions with expressive basis functions, thereby decoupling the lower-level dependence on context. It formalizes the relationship between CSBO and standard SBO and makes it possible to achieve $\(\epsilon\)$-stationarity with near-optimal sample complexity
$\(\tilde{\mathcal{O}}(\epsilon^{-3})\)$. Experiments on inverse problems and hyperparameter tuning demonstrate the effectiveness of the method.

**Strengths:**

1. The paper proposes a reduction framework for CSBO that leverages the efficiency of existing SBO algorithms and rigorously establishes the relationship between CSBO and SBO.

2. The presentation is clear and well structured.

**Weaknesses:**

1. The theoretical analysis hinges on a $\mu$-strongly convex lower-level problem.

2. The treatment of high-dimensional scenarios and alternative basis functions(e.g., neural networks) is limited.

**Questions:**

1. Although the paper achieves the near-optimal sample complexity $\tilde{\mathcal{O}}(\epsilon^{-3})$, “Drawback 1” noted around line 80 appears unresolved. Please clarify what gap remains, whether it is fundamental or technical, and how the reduction might be extended to close it.

2. The experiments compare primarily against \textsc{StocBiO}. Why were other SBO algorithms excluded? Is it straightforward to plug your reduction into alternative SBO solvers (e.g., variance-reduced or momentum-based methods)? A minimal example illustrating such integration would be helpful.

3. A minor suggestion: please use \citet{} and \citep{} appropriately to improve readability.

4. See Weaknesses.

---

> ### Author Response · Authors · 2025-11-21
>
> Thank you for your positive evaluation and constructive feedback.
>
> **1. Remaining gap**
>
> The $\tilde{O}(\epsilon^{-1})$ gap in the bullet point about existing CSBO algorithms refers to the gap between the state-of-the-art $\tilde{O}(\epsilon^{-4})$ complexity for CSBO compared to SBO's $O(\epsilon^{-3})$. Our framework reduces this gap to $\tilde{O}(\text{Poly}(M_\Phi(\epsilon), 1/m_\Phi(\epsilon)))$, which simplifies to $\tilde{O}(1)$ under the conditions of Theorem 4.5. The remaining logarithmic factors arise from the unbounded growth in the number of basis functions required as the lower-level approximation tolerance $\epsilon$ decreases. In our current analysis, achieving the optimal $O(\epsilon^{-3})$ rate would require $M_\Phi(\epsilon)$ and $1/m_\Phi(\epsilon)$ to be uniformly bounded by a constant, which is unlikely to hold for general $f$ and $g$. Still, we match SBO's rate up to logarithmic factors, effectively closing the practical gap.
>
> **2. Alternative SBO solvers**
>
> Our reduction is indeed solver-agnostic. We focused on stocBiO in our experiments since its implementation is publicly available with proven performance on the MNIST hyper-cleaning benchmark, which makes it a reliable and reproducible backbone for our experiments. Re-implementing each algorithm, particularly those with nontrivial hyperparameter schedules and lacking reproducible code, is delicate and risks yielding unfairly weak baselines. To support that our conclusions are not specific to stocBiO, we ran additional experiments [see additional experiment \#2 in the comments to reviewer YLFf] using $\texttt{SOBA}$ and $\texttt{SUSTAIN}$ as backbones. Here, the parametrization consistently improves both training and validation losses across all solvers, indicating that the advantages of our parametrization are not tied to a particular choice of solver.
>
> **3. Citation formatting**
>
> Thank you for the suggestion. We will properly use these commands to improve readability.
>
> **4.1 Strong convexity assumption**
>
> We acknowledge that this standard assumption has limitations and that recent works have notably relaxed the lower-level assumptions to weaker ones, such as the PL condition. However, CSBO introduces substantial challenges to SBO, and the absence of a unique lower-level solution would significantly complicate the analysis. To the best of our knowledge, all existing CSBO methods either assume strong convexity of the lower-level objective or impose a specific structure that guarantees uniqueness of the lower-level solution. For these reasons, we believe that assuming a strongly convex lower-level is appropriate as a first step in improving CSBO complexity guarantees.
>
> **4.2 High-dimensional experiments**
>
> We want to emphasize that, despite the simplicity of MNIST as a lower-level, the data-cleaning task is non-trivial due to the dimension of the upper level ($d_x = N_\text{train} = 19,000$). For this reason, MNIST data-cleaning is the most prevalent benchmark in SBO. We conducted a larger-scale experiment with a non-convex lower level in which our reduction remains effective [see additional experiment \#1 in the comments to reviewer YLFf].

---

### Official Review · Reviewer_7i2v · 2025-10-31

**Soundness:** 3
**Presentation:** 3
**Contribution:** 3
**Rating:** 6
**Confidence:** 4

**Summary:**

This paper studied the contextual stochastic bilevel optimization and proposed a novel method by approximating the lower-level solution using expressive basis functions. In this way, the contextual stochastic bilevel optimization problem is transferred back to standard stochastic bilevel optimization problem and can be solved by standard bilevel methods with enhanced convergence rate. This paper provided the sufficient conditions to make the approximation errors from the basis expression small and demonstrated the Chebyshev polynomial satisfies those conditions. Numerical experiments verify the effectiveness of the proposed methods.

**Strengths:**

1. Using an expressive basis to represent the context space is novel and effective, especially when the space is finite, as in predominate contextual stochastic optimization applications like meta-learning.
2. This paper also offers two effective basis: Chebyshev and Fourier polynomial, which has been studied in approximation theory but have not be leveraged into bilevel optimization. The strategy has also been proved effective in experiments.
3. The representation improves the convergence rate of existing literature on contextual bilevel optimization and matches that of stochastic bilevel optimization.

**Weaknesses:**

1. The proposed expressive basis seems only effective when the dimension of contextual variable is finite or has bounded support with non-vanishing density. But the authors also claim that meta-learning and distributionally robust optimization on the compact sets satisfy these conditions.
2. It would be valuable to quantify the error introduced by vanishing densities that violate the assumption, to indicate how sensitive the method is on general contextual bilevel problems.
3. In theory, how should $N_\Phi$ scale with $\epsilon$? The analysis suggests $N_\Phi$ should be large. But in practice, $N=3$ gives a satisfactory result. Does that mean the analysis is not tight for problems with special structures? Could you comment on what determines $N$ and when theory predicts that small $N$ is enough?

**Questions:**

1. In Line 281, I don’t see why a small minimum eigenvalue implies a low-dimensional subspace. Shouldn’t the effective dimension be driven by the number of large eigenvalues instead?
2. Numerical complexity of calculating the basis weight matrix when dimension is high.
3. Missing references on first-order and variance reduction based bilevel methods:

[1] On penalty-based bilevel gradient descent method. H Shen, T Chen. ICML 2023.

[2] A framework for bilevel optimization that enables stochastic and global variance reduction algorithms. Mathieu Dagréou, Pierre Ablin, Samuel Vaiter, Thomas Moreau. NeurIPS 2022.

[3] Provably Faster Algorithms for Bilevel Optimization. Junjie Yang, Kaiyi Ji, Yingbin Liang. NeurIPS 2021.

---

> ### Author Response · Authors · 2025-11-21
>
> Thank you for your supportive review and insightful questions. We appreciate your recognition of our contributions.
>
> **Robustness to assumption violations**
>
> We want to emphasize that, in practice, expressive bases can be efficient even when several of our assumptions are violated [see additional experiment \#1 in the comments to reviewer YLFf]. The conditions of Theorem 4.5 are *sufficient* (not necessary) to obtain guarantees for the particular choice of Chebyshev polynomials, and the bounds we derive are stated for general functions $f$ and $g$. One can obtain much stronger results if the structure of $y^\star$ is known (e.g. if $y^\star$ is affine in $\xi$ then $\Phi(\xi) = (1, \xi)$ can approximate $y^\star$ exactly so that $M_\Phi(\epsilon)$ and $1/m_\Phi(\epsilon)$ are upper bounded by small constants). Finally, the ``non-vanishing density" is a conservative requirement in condition c.1, which ensures that the Chebyshev polynomials are not perfectly correlated in expectation. For example, if $P_\xi = \delta_0$, then $T_1(\xi) = \xi$ and $T_3(\xi) = 4\xi^3 - 3\xi$ coincide on the support of $\xi$, hence $\lambda_\text{min}(\Sigma_\Phi) = 0$. In practice, many distributions whose densities vanish on subsets of $\Xi$ still yield well-conditioned Chebyshev polynomials even though they fall outside the conditions of Theorem 4.5.
>
> **Minimum eigenvalue clarification**
>
> You are absolutely correct, our phrasing was imprecise. A more accurate statement is that a small minimum eigenvalue indicates that the feature vectors are concentrated near a *lower*-dimensional subspace, not in a low-dimensional subspace. Since strong convexity of the lower level in the reformulation requires the covariance matrix to be full rank, $m_\Phi(\epsilon)$ precisely captures the non-degeneracy of the features needed for our analysis. We have revised this statement for accuracy.
>
> **Complexity of the lower-level in high dimension**
>
> The key insight is that sample complexity depends on $N$ only through the reformulation's regularities (that worsen with $N$), which we bound via $M_\Phi$ and $m_\Phi$. While we express $M_\Phi$ and $m_\Phi$ w.r.t. $\epsilon$ since it is more natural for our analysis, they can equivalently be expressed w.r.t. $N$ via the bijection $N_\Phi(\epsilon)$. Recall that $G_{\Phi^{[N]}}(x, W)$ is the lower-level objective of the reformulation using $N$ basis functions.
> For a $\mu_G$-strongly convex and $L_G$-smooth function, the sample complexity to achieve $\mathbb{E}[\|W - W^\star\|^2] \leq \epsilon^2$ with SGD is in the order of $\tilde{O}(\sigma_G^2/(\mu_G \cdot \epsilon^2))$ where $\sigma_G^2$ is the variance of $\nabla_2 G$ and $\mu_G$ is the strong convexity constant of $G$ w.r.t. $W$.
> We show in our proof that $\sigma_G^2 = O(M_\Phi^4(N))$ (Lemma C.7) and $\mu_G \geq \mu \cdot m_\Phi(N)$ (Lemma C.2), so that the sample complexity of the lower level is $\tilde{O}(\epsilon^{-2} \cdot M_\Phi^4(N)/m_\Phi(N))$ and collapses to $\tilde{O}(\epsilon^{-2})$ under conditions c.1-3.
>
> **Missing references** Thank you for these. While [3] is already cited, [1] and [2] are important references that we had inadvertently omitted. We have incorporated them in the revised version.

---

### Official Review · Reviewer_dYLE · 2025-11-04

**Soundness:** 2
**Presentation:** 2
**Contribution:** 3
**Rating:** 4
**Confidence:** 3

**Summary:**

This paper studies a contextual stochastic bilevel optimization (SBO) problem, where the lower-level objective depends on both a random context and the upper-level variable. Compared to standard SBO, the main challenge lies in estimating the hypergradient: a naive approach requires solving multiple lower-level problems and sampling from a conditional distribution for arbitrary contexts. To address these challenges, the authors map the context to a feature space via Chebyshev polynomial bases, and parametrize the lower-level solution as a linear function of the resulting feature vector. This reformulation converts the contextual SBO into a standard SBO that avoids the need for conditional sampling. The paper further identifies conditions under which the transformed problem retains the regularity of the original one and demonstrates how to construct an approximate solution to the contextual SBO from the standard SBO solution.

**Strengths:**

- **Originality**: This paper introduces tools from approximation theory to tackle contextual SBO. To the best of my knowledge, this has not been explored in the existing literature.
- **Significance**: The proposed method yields a simple reformulation of the original problem and eliminates the need for a conditional sampling oracle. Under suitable assumptions, it also improves the sample complexity for finding a stationary solution of the context SBO from the previously known $\tilde{O}(\epsilon^{-4})$ to $\tilde{O}(\epsilon^{-3})$.

**Weaknesses:**

- A central question for the proposed approach is how accurately the reformulated SBO problem approximates the original contextual SBO. In my view, this point is not clearly articulated in the paper; instead, much of the technical complexity is deferred to the notion of an expressive basis (Definition 3.4). Specifically, Definition 3.4 assumes that the approximation error can be made arbitrarily small by increasing the number of basis functions, but it is unclear to me whether this is always achievable in practice. Intuitively, to parametrize the lower-level solution $y^*(x,\xi)$ as a linear map of the context features, some structural assumptions on the lower-level objective are needed, yet such assumptions are not explicitly discussed in the paper.
- Another concern is that, after reformulation, the dimensionality of the lower-level subproblem increases from $d_y$ to $d_y \times N_{\Phi}(\epsilon)$. To achieve a faithful approximation of the original contextual problem, $N_{\Phi}(\epsilon)$ may need to be large, potentially leading to significant computational overhead. In such cases, the reformulated approach may not offer a clear advantage over directly solving multiple lower-level subproblems.

**Questions:**

In light of the weaknesses discussed above, I have the following clarifying questions:
- The authors suggest that the assumptions in Theorem 4.5 ensure the Cheyshev polynomial basis is expressive. In particular, they assume the function $G(x,y,\xi)$ is analytic in $(y, \xi)$. Could the authors elaborate on how restrictive this assumption is? For instance, in my understanding, an analytic function is typically defined for continuous variables, so this assumption will not hold when the context space $\Xi$ is discrete.
- Could the authors provide a more quantitative estimate of the feature-space dimension $N_{\Phi}(\epsilon)$? Specifically, how does it scale with other problem parameters?
- I also find the definitions of feature map and expressive basis in Definitions 3.3 and 3.4 somewhat dense. To improve readability, it would be helpful to include concrete examples that illustrate these definitions.

---

> ### Author Response · Authors · 2025-11-21
>
> Thank you for your thoughtful feedback. We address your main concerns below.
>
> **Concrete example**
>
> We propose here an example that could improve readability regarding Definitions 3.3 and 3.4 as well as $N_\Phi(\epsilon)$, $M_\Phi(\epsilon)$, and $m_\Phi(\epsilon)$.
>
> Consider a CSBO problem such that $y^\star(x, \xi) = \cos(\xi\cdot \cos(x))$ and $P_\xi = U(0,1)$. Let $\Phi$ be the polynomial basis $\Phi = (1,\xi, \xi^2, ...)$. Then $\Phi^{[N]} \triangleq (1,...,\xi^{N-1})$ and the Taylor series of $y^\star$ at $\xi = 0$ gives for $N$ even and $\xi \in [0,1]$: $\left|y^\star(x, \xi) - W(x)\Phi^{[N]}(\xi)\right| \leq \frac{|cos(x)\xi|^{N}}{N!} \leq \frac{1}{N!}$. To guarantee a uniform error of at most $\epsilon$, it suffices to take $N_\Phi(\epsilon) = O(\ln \epsilon^{-1})$. By definition, we have $\Phi^\epsilon \triangleq \Phi^{[N_\Phi(\epsilon)]} =  (1,...,\xi^{N_\Phi(\epsilon)-1})$ and the covariance matrix $\Sigma_{\Phi^\epsilon} \triangleq \mathbb{E}[\Phi^\epsilon\Phi^{\epsilon\top}]$ is the Hilbert matrix. It follows that $M_\Phi(\epsilon) = \sqrt{N_\Phi(\epsilon)}$ and $m_\Phi(\epsilon) = \lambda_\text{min}(\Sigma_{\Phi^\epsilon})$.
>
> **Scaling of $N_\Phi(\epsilon)$ and reformulation complexity**
>
> The number of basis functions $N_\Phi(\epsilon)$ is an intuitive proxy for the difficulty of the reformulated problem since a larger $N_\Phi(\epsilon)$ indeed results in a higher-dimensional lower-level problem. However, Theorem 4.4 shows that the sample complexity of the reformulation depends on $M_\Phi(\epsilon)$ and $1/m_\Phi(\epsilon)$ (both growing with $N_\Phi(\epsilon)$). In other words, the *effective* complexity of the reformulated problem, including the lower-level dimension ($d_y \times N_\Phi(\epsilon)$), conditioning, and Lipschitz constants, is fully captured by $M_\Phi(\epsilon)$ and $m_\Phi(\epsilon)$, rather than $N_\Phi(\epsilon)$ itself. In fact, a basis with a small $N_\Phi(\epsilon)$ can still induce large Lipschitz constants or an ill-conditioned feature covariance, leading to poor complexity. In contrast, other bases with greater $N_\Phi(\epsilon)$ but moderate $M_\Phi(\epsilon)$ and $m_\Phi(\epsilon)$ can yield a much simpler reformulation. For this reason, we focus our attention on $M_\Phi(\epsilon)$ and $m_\Phi(\epsilon)$, and view $N_\Phi(\epsilon)$ as a latent variable fully captured by $M_\Phi(\epsilon)$ and $m_\Phi(\epsilon)$.
>
> The example above clearly exhibits this phenomenon. Indeed, it can be shown that $\lambda_\text{min}(\Sigma) = \Theta(\sqrt{N_\Phi(\epsilon)}/(1+\sqrt{2})^{4N_\Phi(\epsilon)})$. It follows that $1/m_\Phi(\epsilon) = \Theta(\epsilon^{-c} / \sqrt{\ln \epsilon^{-1}})$ for some $c > 0$. Therefore, despite the growth of $N_\Phi(\epsilon)$ being well controlled, the degeneracy of the basis makes the reformulation very weakly strongly convex, which results in sampling guarantees $\tilde{O}(\epsilon^{-3-c})$ worse than $\tilde{O}(\epsilon^{-3})$. However, the conditions c.1-3 hold, and the Chebyshev basis achieves near-optimal sample complexity $\tilde{O}(\epsilon^{-3})$.
>
> The table below clarifies how our method relates to existing work:
> | | Sampling Oracle | SBO with $m$ subproblems | CSBO | CSBO + condition c.1-3
> |---|---|---|---|---|
> | Guo et al. [6] | Conditional | $O(m\epsilon^{-3})$ | X | X |
> | Hu et al. [1] | Conditional | $\tilde{O}(\epsilon^{-4})$ | $\tilde{O}(\epsilon^{-4})$ | $\tilde{O}(\epsilon^{-4})$ |
> | Ours | Joint only | $\tilde{O}(\epsilon^{-3})$ | $\tilde{O}\left(\epsilon^{-3} \cdot \text{poly}\left(M_\Phi(\epsilon), \frac{1}{m_\Phi(\epsilon)}\right)\right)$ | $\tilde{O}(\epsilon^{-3})$ |
>
>
> **Expressiveness**
>
> Definition 3.4 keeps our framework intentionally general: it does not impose structural restrictions on how $y^\star(x,\xi)$ depends on $\xi$, allowing expressive bases to arise in many settings beyond those we analyze. Theorem 4.5 provides one concrete set of *sufficient* conditions satisfied by many ML applications under which Chebyshev polynomials yield strong complexity guarantees. Importantly, expressive bases may still exist even when $y^\star(x,\xi)$ is discontinuous in $\xi$ and when $\Xi$ is unbounded. For example, $y^\star(x,\xi) = x \cdot 1_{\{\lfloor \xi \rfloor \text{ is odd}\}}$ is exactly representable over $\Xi = \mathbb{R}$ with a single basis function $\Phi = \{1_{\{\lfloor \xi \rfloor \text{ is odd}\}}\}$.
>
> **Assumptions in Theorem 4.5**
>
> When $\Xi$ is discrete, the lower-level objective $G$ can be analytically continued over the convex hull of $\Xi$, so conditions c.1-3 hold and Chebyshev polynomials provide near-optimal guarantees.
>
> In the uncommon case where $G$ is continuous in $\xi$ but not analytic, the provable *uniform* convergence rate of Chebyshev polynomials is only algebraic, giving weaker guarantees *under a general distribution* $\mathbb{P}_\xi$. Still, as long as the approximation error of a basis $\Phi$ is small *in expectation* w.r.t. $\xi$ (Definition 3.4), Theorem 4.4 and its sample guarantees hold.

---

### Author Response · Authors · 2025-12-03

Dear Area Chair,

Given the exceptional circumstances surrounding this year's ICLR and the modifications to the review process, we thank you for your efforts and recognize that these changes may result in an increased workload. Below, we summarize the contributions of our paper and explain how we addressed the main concerns of the reviewers. To support these points, we conducted experiments that further demonstrate the value and broad applicability of our work.

Our main contribution is a novel reduction framework for CSBO that parametrizes lower-level solutions via basis functions, resulting in a standard SBO problem. This reduction enables tackling CSBO with existing SBO solvers, and eliminates the need for impractical conditional sampling oracles required by prior CSBO methods. We establish the sample complexity bound $\tilde{\mathcal O}\left(\epsilon^{-3}\text{poly}(M_\Phi(\epsilon), m_\Phi^{-1}(\epsilon))\right)$ that depends on the basis $\Phi$ and context distribution $\mathbb P_\xi$. Under mild conditions satisfied in many ML applications, the specific choice of Chebyshev polynomials as basis yields near-optimal sample complexity $\tilde{\mathcal O}(\epsilon^{-3})$ matching standard SBO rates up to logarithmic factors and strictly improves the best-known complexity rate for CSBO. Finally, we provide empirical evidence of its effectiveness on inverse and hyperparameter optimization tasks.

Reviewer **dYLE**'s main concerns were (i) insufficient clarity around Definition 3.4, (ii) potential computational overhead due to the increased lower-level dimension $N_\Phi(\epsilon)$, and (iii) a misunderstanding of how restrictive the analyticity assumption is. We addressed these points by adding a concrete illustrative example immediately following Definition 3.4 to clarify the notion of expressiveness and the relationships between $N_\Phi(\epsilon)$, $M_\Phi(\epsilon)$, and $m_\Phi(\epsilon)$. We then explained that the complexity resulting from enlarging the lower-level dimension by $N_\Phi(\epsilon)$ is fully captured by the metrics $M_\Phi(\epsilon)$ and $m_\Phi(\epsilon)$. Finally, we clarified that when contexts are discrete, the analyticity requirement is not restrictive and is automatically satisfied via analytic continuation, contrary to the reviewer’s interpretation that analyticity would fail to hold in this setting.

Reviewer **7i2v** expressed concern that the method might only be effective for discrete contexts or contexts with vanishing densities, and questioned the practical versus theoretical scaling of the basis size $N_\Phi(\epsilon)$. We clarified that the assumptions in Theorem 4.5 are conservative sufficient conditions, not necessary ones. Even when these conditions do not hold, the more general guarantees of Theorem 4.4 still apply, and Chebyshev bases can still yield near-optimal sample complexity. To support this, we conducted Additional Experiment #1 on a larger scale and non-convex lower-level problem, which demonstrates that our approach remains efficient in settings where most theoretical assumptions are violated.

Reviewer **yLsY** noted an "unresolved" theoretical gap between CSBO and SBO convergence rates, and they expressed a desire for larger-scale experiments as well as results using additional backbone SBO solvers. We clarified that, although a $\tilde{\mathcal O}(1)$ gap remains between CSBO and SBO, our reduction improves the $\tilde{\mathcal O}(\epsilon^{-1})$ gap from prior works to $\widetilde{\mathcal O}(1)$ and effectively closes the practical gap. We addressed scalability via Additional Experiment #1, and demonstrated the solver-agnostic nature of our reduction via Additional Experiment #2, where the alternative SBO backbones SOBA and SUSTAIN show consistent improvements and confirm that our framework integrates seamlessly with various SBO solvers.

Reviewer **YLFf**’s primary concern stemmed from a misunderstanding of Condition c.1 in Theorem 4.5, which they interpreted as restricting the scope of this result to discrete contexts (i.e., effectively an SBO formulation with multiple subproblems). We pointed out that c.1 is an “OR” condition that explicitly includes continuous context distributions, and thus generalizes prior work while eliminating the need for conditional sampling (as detailed in the accompanying table). We also clarified that, unlike standard MNIST classification, MNIST data cleaning is a non-trivial task and a widely used benchmark in SBO. We further addressed their concerns about scalability and baselines through Additional Experiments #1 and #2, where our approach outperforms the baselines on a larger-scale problem and across multiple SBO backbone solvers. Finally, we emphasized the relevance of the CSBO formulation for ML applications and clarified that adaptive bases were intentionally excluded because they offer weaker theoretical guarantees.

We hope this helps you make an informed assessment of our work.

Thank you,

The Authors

---

### Meta-Review · Area_Chair_4NJp · 2026-01-05

**Summary:**

The authors propose a novel reduction framework for Contextual Stochastic Bilevel Optimization (CSBO). By parametrizing the context-specific lower-level solutions via basis functions (e.g., Chebyshev polynomials), this framework decouples the lower-level dependence on context and transforms the CSBO problem into a standard SBO problem1. This approach eliminates the need for impractical conditional sampling oracles and theoretically achieves a near-optimal sample complexity of $\tilde{\mathcal{O}}(\epsilon^{-3})$.

The reviewers' sentiments were divided (scores 4, 6, 6, 2). Reviewers 7i2v and yLsY positively evaluated the framework's novelty and its ability to solve CSBO using strictly joint samples. Reviewer dYLE gave a borderline score (4), concerned that the parametrization might increase computational overhead as the number of basis functions grows. In response, the authors clarified that the complexity is primarily driven by the basis conditioning ($m_{\Phi}$) rather than the raw dimension ($N_{\Phi}$), and that efficient bases like Chebyshev polynomials maintain favorable properties even in high dimensions. Reviewer YLFf assigned a rejection score (2), primarily arguing that the theoretical conditions (specifically c.1) were too restrictive and the experimental baselines were insufficient. The authors forcefully addressed this by demonstrating that the reviewer misinterpreted condition c.1, which explicitly covers continuous contexts, and by providing comprehensive additional experiments on a large-scale non-convex problem (CIFAR-100) and with additional solvers (SOBA, SUSTAIN). Overall, the authors' rebuttal was highly effective, successfully resolving key theoretical misunderstandings and substantially strengthening the empirical evidence against the main criticisms.

The authors provided a compelling rebuttal that addressed the primary grounds for rejection. Specifically, they clarified Reviewer YLFf's misinterpretation of condition c.1, demonstrating that their reduction framework explicitly covers continuous context distributions . Furthermore, the authors conducted additional large-scale experiments on a non-convex problem (CIFAR-100, ResNet-18) and integrated additional solvers (SOBA, SUSTAIN), proving that the parametrization remains effective and scalable even when theoretical assumptions are violated . Given that the paper effectively bridges the gap between CSBO and SBO with strong theoretical and empirical backing, it merits acceptance.

**Reviewer Concerns:**

**Addressed**:

The authors forcefully corrected Reviewer YLFf's misunderstanding regarding condition c.1. Addressing the reviewer's interpretation that the condition applied only to discrete contexts, the authors clarified that it is an "OR" condition that explicitly covers continuous context distributions with non-vanishing density, thereby establishing a clear theoretical advantage over [1].

Addressing concerns about computational overhead (Reviewer dYLE) and "toy problem" baselines (Reviewer YLFf), the authors added convincing large-scale experiments (CIFAR-100, ResNet-18). These experiments demonstrated the method's scalability in high-dimensional settings (lower-level parameters > 280k) and proved its effectiveness even in non-convex scenarios where theoretical assumptions (like strong convexity) are violated.

 In response to criticisms regarding limited baselines from Reviewers yLsY and YLFf, the authors integrated comparisons with additional SBO solvers (SOBA and SUSTAIN) during the rebuttal. The results showed consistent performance improvements across different backbones, demonstrating the solver-agnostic nature of the reduction framework.

 The authors clarified to Reviewer dYLE that complexity is not driven solely by the number of basis functions $N$, but by their conditioning ($M_{\Phi}, m_{\Phi}$). Using concrete examples, they argued that Chebyshev bases maintain favorable properties even as $N$ increases, dispelling concerns about dimensional explosion.

**Outstanding**:

 Reviewer YLFf suggested using "learned basis functions" instead of fixed ones for better approximation. While the authors acknowledged this as a promising future direction, they maintained the use of fixed bases in this work to preserve theoretical rigor regarding convexity and convergence analysis . This remains a limitation rather than a fatal flaw.

 As confirmed in the response to Reviewer yLsY, a logarithmic gap $\tilde{\mathcal{O}}(1)$ in sample complexity remains between CSBO and standard SBO (due to the growth of basis functions). While the authors argue this impact is negligible in practice, the theoretical gap has not been completely closed.

**Ref:**

[1]Zhishuai Guo, Quanqi Hu, Lijun Zhang, and Tianbao Yang. Randomized stochastic variance-reduced methods for multi-task stochastic bilevel optimization. arXiv preprint arXiv:2105.02266, 2021.

**Reviewer Scores:**

**Reviewer dYLE (Score: 4  -> Est. unchanged):**

Reviewer dYLE would likely keep the same overall score (4: marginally below acceptance) or raise it marginally. Their main concerns centered on the clarity of definitions regarding basis expressiveness and the potential computational overhead as the number of basis functions increases. In the rebuttal, the authors provided a concrete mathematical example and clarified that the complexity of the reformulated problem depends primarily on the conditioning of the basis functions ($m_{\Phi}$) rather than solely on their dimension ($N_{\Phi}$). While this theoretical clarification was strong and addressed the question of vague definitions, the reviewer might still retain some caution regarding the practical trade-offs of the reformulation in high-dimensional settings without seeing code.

**Reviewer 7i2v (Score: 6  -> Est. unchanged):**

Reviewer 7i2v would likely keep the same overall score (6: Marginally above acceptance). This reviewer was generally supportive but expressed concern about the method's robustness when theoretical assumptions, such as non-vanishing density, are violated in practice. To address this, the authors added an experiment with a non-convex ResNet model, demonstrating that the method remains effective even when these assumptions are not met.

**Reviewer yLsY (Score: 6  -> Est. unchanged):**

Reviewer yLsY would likely keep the same overall score (6: Marginally above acceptance). This reviewer made a very specific and actionable request: they wanted to see the method integrated with other SBO solvers (beyond `stocBiO`) to prove its generality. The authors met this requirement by presenting experimental results integrated with SOBA and SUSTAIN solvers in the rebuttal, showing consistent performance improvements.

**Reviewer YLFf (Score: 2  -> Est. 4):**

Reviewer YLFf would likely increase their score to 4 (Marginally below acceptance). This low score stemmed primarily from a core theoretical misunderstanding, where the reviewer incorrectly believed theorem condition c.1 applied only to discrete contexts. The authors explicitly corrected this, pointing out that the condition covers continuous context distributions, thereby overturning the main basis for rejection. However, the reviewer also expressed dissatisfaction with the scale of experiments and the lack of "learned basis functions". Although the authors added a large-scale experiment, the issue of learned basis functions was relegated to future work.

---

### Decision · Program_Chairs · 2026-01-26

Accept (Poster)